# What About Taking Policy as Input of Value Function: Policy-extended Value Function Approximator

## Abstract

The value function lies in the heart of Reinforcement Learning (RL), which defines the long-term evaluation of a policy in a given state. In this paper, we propose Policy-extended Value Function Approximator (PeVFA) which extends the conventional value to be not only a function of state but also an explicit policy representation. Such an extension enables PeVFA to preserve values of multiple policies in contrast to a conventional one with limited capacity for only one policy, inducing the new characteristic of *value generalization among policies*. From both the theoretical and empirical lens, we study value generalization along the policy improvement path (called local generalization), from which we derive a new form of Generalized Policy Iteration with PeVFA to improve the conventional learning process. Besides, we propose a framework to learn the representation of an RL policy, studying several different approaches to learn an effective policy representation from policy network parameters and state-action pairs through contrastive learning and action prediction. In our experiments, Proximal Policy Optimization (PPO) with PeVFA significantly outperforms its vanilla counterpart in MuJoCo continuous control tasks, demonstrating the effectiveness of value generalization offered by PeVFA and policy representation learning.

## 1 Introduction

Reinforcement learning (RL) has been widely considered as a promising way to learn optimal policies in many decision making problems (Mnih et al., 2015; Lillicrap et al., 2015; Silver et al., 2016; You et al., 2018; Schreck et al., 2019; Vinyals et al., 2019; Hafner et al., 2020). Lying in the heart of RL is the value function which defines the long-term evaluation of a policy. With function approximation (e.g., deep neural networks), a value function approximator (VFA) is able to approximate the values of a policy under large and continuous state spaces. As commonly recognized, most RL algorithms can be described as Generalized Policy Iteration (GPI) (Sutton & Barto, 1998). As illustrated in the left of Figure 1, at each iteration the VFA is trained to approximate the true values of current policy, regarding which the policy are improved. However, value approximation can never be perfect and its quality influences the effectiveness of policy improvement, thus raising a requirement for better value approximation (v. Hasselt, 2010; Bellemare et al., 2017; Fujimoto et al., 2018).

Since a conventional VFA only approximates the values (i.e., knowledge (Sutton et al., 2011)) for one policy, the knowledge learned from previously encountered policies is not preserved and utilized for future learning in an explicit way. For example in GPI, a conventional VFA cannot track the values of the changing policy by itself and has no idea of the direction of value generalization when approximating the values of a new policy. In this paper, we propose Policy-extended Value Function Approximator (PeVFA), which additionally takes an explicit policy representation as input in contrast to conventional VFA. PeVFA is able to preserve values for multiple policies and induces an appealing characteristic, i.e., value generalization among policies. We study the formal generalization and contraction conditions on the value approximation error of PeVFA, focusing specifically on value generalization along the policy improvement path which we call local generalization. Based on both theoretical and empirical evidences, we propose a new form of GPI with PeVFA (the right of Figure 1) which can benefit from the closer approximation distance induced by local value generalization under some conditions; thus, GPI with PeVFA is expected to be more efficient in consecutive value approximation along the policy improvement path.

Figure 1: Generalized Policy Iteration (GPI) with function approximation. *Left*: GPI with conventional value function approximator $V_\phi^\pi$. *Right*: GPI with PeVFA $\mathbb{V}_\theta(\chi_\pi)$ (Sec. 3) where extra generalization steps exist. The subscripts of policy $\pi$ and value function parameters $\phi, \theta$ denote the iteration number. The squiggle lines represent non-perfect approximation of true values.

Moreover, we propose a framework to learn effective policy representation for an RL policy from policy network parameters and state-action pairs alternatively, through contrastive learning and an auxiliary loss of action prediction. Finally, based on Proximal Policy Optimization (PPO), we derive a practical RL algorithm PPO-PeVFA from the above methods. Our experimental results demonstrate the effectiveness of both value generalization offered by PeVFA and policy representation learning. Our main contributions are summarized as follows:

- We propose PeVFA which improves generalization of values among policies and provide a theoretical analysis of generalization especially in local generalization scenario.

- We propose a new form of GPI with PeVFA resulting in closer value approximation along the policy improvement path demonstrated through experiments.

- To our knowledge, we are the first to learn a representation (low-dimensional embedding) for an RL policy from its network parameters (i.e., weights and biases).

## 2 BACKGROUND

### 2.1 REINFORCEMENT LEARNING

We consider a Markov Decision Process (MDP) defined as $\langle \mathcal{S}, \mathcal{A}, r, \mathcal{P}, \gamma \rangle$ where $\mathcal{S}$ is the state space, $\mathcal{A}$ is the action space, $r$ is the reward function, $\mathcal{P}$ is the transition function and $\gamma \in [0, 1)$ is the discount factor. The goal of an RL agent is to learn a policy $\pi \in \Pi$ where $\pi(a|s)$ is a distribution of action given state that maximizes the expected long-term discounted return. The *state-value function* $v^\pi(s)$ is defined in terms of the expected discounted return obtained through following the policy $\pi$ from a state $s$: $v^\pi(s) = \mathbb{E}_\pi \left[ \sum_{t=0}^\infty \gamma^t r_{t+1} | s_0 = s \right]$ for all $s \in \mathcal{S}$ where $r_{t+1} = r(s_t, a_t)$. We use $V^\pi = v^\pi(\cdot)$ to denote the vector of values for all possible states. Value function is determined by policy $\pi$ and environment models (i.e., $\mathcal{P}$ and $r$). For a conventional value function, policy is modeled implicitly within a table or a function approximator, i.e., a mapping from only state to value. One can refer to Appendix E.1 to see a more detailed description.

### 2.2 EXTENSIONS OF CONVENTIONAL VALUE FUNCTION

Schaul et al. (2015) introduced Universal Value Function Approximators (UVFA) that generalize values over goals in goal-conditioned RL. Similar ideas are also adopted for low-level learning of Hierarchical RL (Nachum et al., 2018). Such extensions are also studied in more challenging RL problems, e.g., opponent modeling (He & Boyd-Graber, 2016; Grover et al., 2018; Tacchetti et al., 2019), and context-based Meta-RL (Rakelly et al., 2019; Lee et al., 2020). General value function (GVF) in (Sutton et al., 2011) are proposed as a form of general knowledge representation through cumulants instead of rewards. In an unified view, each approach generalizes different aspects of the conventional VFA, focusing on different components of the vector form *Bellman equation* (Sutton & Barto, 1998) expanded on as discussed in Appendix E.2.

Concurrent to our work, several works also study to take policy as an input of value functions. Harb et al. (2020) propose Policy Evaluation Networks (PVN) to approximate the objective function $J(\pi) = \mathbb{E}_{s_0 \sim \rho_0} [v^\pi(s_0)]$ of different policy $\pi$, where $\rho_0$ is the initial state distribution. Later

in (Raileanu et al., 2020), Policy-Dynamics Value Function (PDVF) is proposed which takes both policy and task context as additional inputs, for the purpose of value generalization among policies and tasks so that to adapt quickly to new tasks. PDVF can be viewed as an integration of PVN and task-specific context learning (Rakelly et al., 2019; Zhou et al., 2019a). Both PVN and PDVF conduct value approximation for a given collection of policies and then optimize policy with gradients through policy-specific inputs (shorted as *GTPI* below) of well trained value function variants in a zero-shot manner. We view this as a typical case of global generalization discussed further in Sec. 3. In contrast, we focus more on a local generalization scenario and utilize value generalization to improve learning during standard GPI process with no prior policies given.

Closely related to our work, Faccio & Schmidhuber (2020) propose a class of Parameter-based Value Functions (PVFs) which take policy parameters as inputs. Based on PVFs, new policy gradients are introduced in the form of a combination of conventional policy gradients plus GTPI (i.e., by backpropagating through policy parameters in PVFs). Beyond zero-shot policy optimization, PVFs also utilize value generalization of PVFs in the iterative learning process. Our work differs with PVFs at two aspects: first, we consider a general policy representation in our proposed PeVFA along with a framework of policy representation learning, in contrast to parsing the policy parameters directly; second, we do not resort to GTPI for the policy update in our algorithms and only utilize value generalization for more efficient value estimation in GPI. Moreover, in these previous works, it is not clear how the value generalization among policies can be and whether it is beneficial for learning or not. In this paper, we study the condition of beneficial value generalization from both theoretical and empirical lens.

## 3 POLICY-EXTEND VALUE FUNCTION APPROXIMATOR

In this paper, we propose Policy-extended Value Function Approximator (PeVFA), an extension of the conventional value function that explicitly takes policy (representation) as input. Formally, consider a function $g : \Pi \to \mathcal{X} = \mathbb{R}^n$ in a general form that maps any policy $\pi$ to a $n$-dimensional representation $\chi_\pi = g(\pi) \in \mathcal{X}$. The policy-extended value function $\mathbb{V} : \mathcal{S} \times \mathcal{X} \to \mathbb{R}$, defines the values over state and policy space:

$$\mathbb{V}(s, \chi_\pi) = \mathbb{E}_\pi \left[ \sum_{t=0}^{\infty} \gamma^t r_{t+1} | s_0 = s \right], \text{ for all } s \in \mathcal{S}, \pi \in \Pi. \tag{1}$$

When only one policy $\pi$ is considered, $\mathbb{V}(s, \chi_\pi)$ is equivalent in definition to a conventional value function $v^\pi(s)$. Note that if the policy $\pi_\omega$ is explicitly parameterized and its parameters are used as the representation, i.e., $\chi_{\pi_\omega} = \omega$, PeVFA is equivalent to Parameter-based State Value Function (PSVF) in PVF family (Faccio & Schmidhuber, 2020). Similarly, we can define policy-extended action-value function $\mathbb{Q}(s, a, \chi_\pi)$. We use $\mathbb{V}(s, \chi_\pi)$ for demonstration in this paper.

The key point is, $\mathbb{V}(s, \cdot)$ is not defined for a specific policy and is able to preserve the values of multiple policies. With function approximation, a PeVFA is expected to approximate values among policy space, i.e., $\{v^\pi(s)\}_{\pi \in \Pi}$. Formally, with a finite parameter space $\Theta$, we consider the optimal parameter $\theta^*$ with minimum approximation error over all possible states and policies as:

$$\theta^* = \arg\min_{\theta \in \Theta} F(\theta, \Pi), \text{ where } F(\theta, \Pi) = \max_{\pi \in \Pi} \|\mathbb{V}_\theta(\pi) - V^\pi\| = \max_{\pi \in \Pi} \|\mathbb{V}_\theta(\cdot, \pi) - v^\pi(\cdot)\|_\infty,$$

where $F(\theta, \Pi)$ is the overall approximation error of $\mathbb{V}_\theta$ which is defined as the maximum $L_\infty$ norm of value vector distance among $\Pi$, as a typical metric commonly adopted in study on value estimation approximation and policy iteration (Kakade & Langford, 2002; Munos, 2003; Lagoudakis & Parr, 2003). Therefore, the learning process of a PeVFA $\mathbb{V}_\theta$ is $\theta \to \theta^*$.

As commonly recognized, a conventional value function approximator (VFA) can generalize the values to unseen states or state-action pairs after properly training on known states and state-action pairs under a specific policy. Beyond this, PeVFA provides an appealing characteristic that allows values to be generalized among policies, evaluating new policies with the knowledge of old ones. In the following, we study the characteristic of value generalization theoretically (Sec. 3.1) and provide some empirical evidences (Sec. 3.2), from which we finally introduce a new form of GPI (Sec. 3.3).

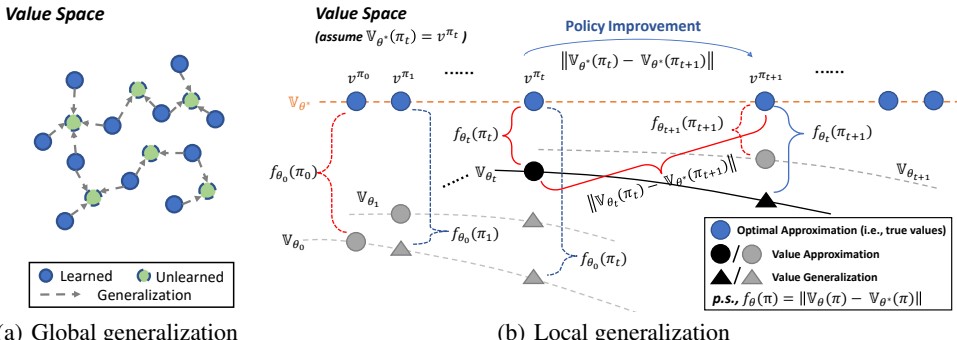

Figure 2: Illustrations of value generalization with PeVFA among policies. Each circle represents the value function of a policy. (a) *Global generalization*: values can generalize to unlearned policies ($\pi' \in \Pi_1$) from already learned policies ($\pi \in \Pi_0$). (b) *Local generalization*: values of previous policies ($\{\pi_i\}_{i=0}^{t}$) can generalize to the successive policy ($\pi_{t+1}$) along policy improvement path.

## 3.1 VALUE GENERALIZATION AMONG POLICIES

Concretely, we introduce two scenarios of value generalization: global generalization and local generalization, which are closely related to many RL problems. An illustration is show in Figure 2. In this section, we first focus on a general two-policy case and study whether the learned value approximation of one policy can generalize to that of the other one. We propose a common form of generalization approximation error (Theorem 1) and analyze the condition of generalization contraction (Corollary 1). Further, we go to the two generalization scenarios as illustrated in Figure 2 and especially introduce the closer approximation distance along the policy improvement path induced by PeVFA (Corollary 2), from which a more efficient form of GPI can be derived (Sec. 3.3).

Formally, consider two policies $\pi_1$ and $\pi_2$ and a PeVFA $\mathbb{V}_\theta$ to approximate the values of them. For convenience of demonstration, we consider an identical representation space (i.e., $\mathcal{X} = \Pi$) and an infinite parameter space $\Theta$ where zero approximation error can be achieved, i.e., $F(\theta^*, \Pi) = 0$. We then define the *approximation loss $f$* of $\mathbb{V}_\theta$ for each policy $\pi \in \Pi$ as $f_\theta(\pi) = \|\mathbb{V}_\theta(\pi) - \mathbb{V}_{\theta^*}(\pi)\| = \|\mathbb{V}_\theta(\cdot, \pi) - \mathbb{V}_{\theta^*}(\cdot, \pi)\|_\infty \geq 0$, which measures the value distance of $\theta$ to the optimal parameter at some policy $\pi$. In the following, we analyze the generalization performance on value estimation of unlearned policy $\pi_2$ when PeVFA $\mathbb{V}_\theta$ is only trained to approximate the values of policy $\pi_1$. We first introduce the two assumptions below:

**Assumption 1** (Contraction). *The learning process $\mathscr{P}$ of value approximation for $\pi_1$ is $\gamma$-contraction, that is $f_{\hat\theta}(\pi_1) \leq \gamma f_\theta(\pi_1)$ where $\gamma \in (0, 1]$ and $\theta \xrightarrow{\mathscr{P}(\pi_1)} \hat\theta$.*

$\theta$ and $\hat\theta$ are the parameters of PeVFA before and after some learning of $\pi_1$'s values. We use $f$ and $\hat{f}$ as abbreviations for $f_\theta$ and $f_{\hat\theta}$ below for the ease of demonstration. Recent works (Keskar et al., 2017; Novak et al., 2018; Wang et al., 2018) suggest that generalization performance is related to local properties of the model. We then assume following smoothness property for following analysis:

**Assumption 2** (Smoothness). *$\hat{f}$ is (or has) Lipschitz continuous (gradient/Hessian) at $\pi_1$ with Lipschitz constant $\hat{L}_0$, e.g., $|\hat{f}(\pi) - \hat{f}(\pi_1)| \leq \hat{L}_0 \cdot d(\pi, \pi_1)$ for $\pi \in \Pi$ with some metric space $(\Pi, d)$.*

With above two assumptions, next we derive a general form of an upper bound of the generalized value approximation for unlearned policy $\pi_2$ (i.e., $\hat{f}(\pi_2)$) as follows:

**Theorem 1.** *Under Assumption 1 and 2, when $f(\pi_1) \leq f(\pi_2)$, we have the following bound:*

$$\hat{f}(\pi_2) \leq \underbrace{\gamma f(\pi_2)}_{\text{generalized contraction}} + \underbrace{\mathcal{M}(\pi_1, \pi_2, \hat{L})}_{\text{locality margin}}. \tag{2}$$

*The form of $\mathcal{M}$ depends on the smoothness property, e.g., $\mathcal{M}(\pi_1, \pi_2, \hat{L}) = \hat{L}_0 \cdot d(\pi, \pi_1)$ when $\hat{f}$ is Lipschitz continuous. See Appendix A.1 for more instances of $\mathcal{M}$ when higher-order smoothness properties (Nesterov & Polyak, 2006) are considered.*

*Proof.* See Appendix A.1. The main idea is to chain the Lipschitz continuity upper bound (Assumption 2), the contraction upper bound (Assumption 1), and the inequality of $f$. □

**Remark 1.** *The case $f(\pi_1) \leq f(\pi_2)$ considered in Theorem 1 can usually exist since only $\pi_1$'s values are learned. Under such circumstances, we suggest that the complementary case $f(\pi_1) > f(\pi_2)$ is acceptable since the approximation error of $\pi_2$ is already lower than the trained one.*

Theorem 1 provides a generalization upper bound for $\hat{f}(\pi_2)$, as a generalized contraction on $f(\pi_2)$ plus a locality margin term $\mathcal{M}$ which is related to $\pi_1$, $\pi_2$ and the smoothness property $\hat{L}$. Further, we analyze the condition when a PeVFA can also obtain a contraction approximation for unlearned policy $\pi_2$ though only trained on $\pi_1$, as below:

**Corollary 1.** *Followed by Theorem 1, consider $\hat{f}$ is Lipschitz continuous and $f(\pi_2) \neq 0$, we have, $\hat{f}(\pi_2) \leq \gamma_g f(\pi_2)$ where $\gamma_g = \gamma + \frac{\hat{L}_0 \cdot d(\pi_1, \pi_2)}{f(\pi_2)}$. When $f(\pi_2) \geq \frac{\hat{L}_0 \cdot d(\pi_1, \pi_2)}{1-\gamma}$, $\theta \xrightarrow{\mathscr{P}(\pi_1)} \hat{\theta}$ is also a $\gamma_g$-contraction for value approximation of $\pi_2$, with $\gamma_g \in (0, 1]$.*

*Proof.* Replace $\mathcal{M}$ with Lipschitz continuous upper bound and transform the RHS in Equation 2, then let RHS not greater than $f(\pi_2)$. See Appendix A.2 for complete derivation. □

**Remark 2.** *From the generalization contraction condition provided in Corollary 1, we can find that: as i. $\gamma \to 0$, or ii. $d(\pi_1, \pi_2) \to 0$, or iii. $\hat{L}_0 \to 0$, the contraction condition is easier to achieve (or the contraction gets tighter), i.e., the generalization on unlearned policy $\pi_2$ is better.*

In the above, we discuss value generalization in a two-policy case, i.e., from one learned policy to another unlearned policy. Global generalization in Figure 2(a) is an extension scenario of the two-policy case where values generalize from a known policy set ($\pi \in \Pi_0$) of which the values are learned to unseen policies ($\pi' \in \Pi_1$). In this paper, we focus more on local generalization of PeVFA as shown in Figure 2(b), Recall the GPI form shown in the left of Figure 1, at each iteration $t$, value generalization from current policy $\pi_t$ to improved policy $\pi_{t+1}$ is exactly the two-policy case we discussed above. However, it is unclear how local generalization can impact the value estimation in GPI (i,e., policy evaluation). To this end, we propose the following Corollary to see a connection between local generalization of PeVFA and a closer value approximation distance:

**Corollary 2.** *At iteration $t$ in local generalization scenario of PeVFA (Figure 2(b)), if $f_{\theta_t}(\pi_t) + f_{\theta_t}(\pi_{t+1}) \leq \|\mathbb{V}_{\theta^*}(\pi_t) - \mathbb{V}_{\theta^*}(\pi_{t+1})\|$, then $f_{\theta_t}(\pi_{t+1}) \leq \|\mathbb{V}_{\theta_t}(\pi_t) - \mathbb{V}_{\theta^*}(\pi_{t+1})\|$.*

*Proof.* Proof can be obtained by applying *Triangle Inequality*. See Appendix A.3. □

Corollary 2 indicates that local generalization of PeVFA can induce a more preferable start point ($\mathbb{V}_{\theta_t}(\pi_{t+1})$) which is closer to the optimal approximation target ($\mathbb{V}_{\theta^*}(\pi_{t+1})$) than the conventional one ($V_{\theta_t}^{\pi_t}$, equivalent to $\mathbb{V}_{\theta_t}(\pi_t)$ in definition) for policy evaluation process at iteration $t + 1$.

**Remark 3.** *With closer start points $\mathbb{V}_{\theta_t}(\pi_{t+1})$, policy evaluation (i.e., minimize $f_{\theta_t(\pi_{t+1})}$) can be more efficient with PeVFA. One can assume an ideal case with perfect local generalization, where policy evaluation is no longer necessary and policy improvement can be performed consecutively.*

### 3.2 EMPIRICAL EVIDENCES FOR TWO KINDS OF GENERALIZATION

Beyond theoretical discussion, we empirically investigate whether two kinds of generalization can be achieved by a PeVFA neural network. First, we demonstrate global generalization in a continuous 2D Point Walker environment. We build the policy sets $\Pi$ with synthetic policies, each of which is a randomly initialized 2-layer *tanh*-activated neural policy network with 2 units for each layer. Here we use the concatenation of all weights and biases of the policy network as representation $\chi_\pi$ for each policy $\pi$ for demonstration, called Raw Policy Representation (RPR), while more advanced policy representation methods will be introduced in Sec. 4. We rollout the policies in the environment to collect trajectories $\mathcal{T} = \{\tau_i\}_{i=0}^k$ and then obtain the dataset $\{(\chi_{\pi_j}, \mathcal{T}_{\pi_j})\}_{j=0}^n$. We separate the synthetic policy set into training set (i.e., known policies $\Pi_0$) and testing set (i.e., unseen policies $\Pi_1$). Figure 3(a) shows that a PeVFA trained with data collected by $\Pi_0$ achieves comparable testing approximation performance when evaluating values of policies in $\Pi_1$ (the average MSE on training/testing set is 2.909 and 4.155), as well as almost maintains the order of optimality among policies. More experimental details are provided in Appendix B.1.

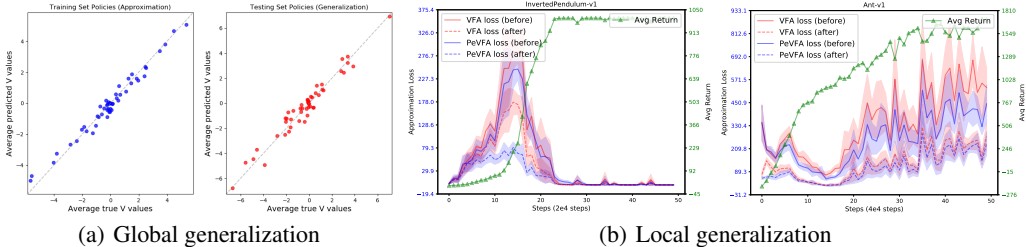

(a) Global generalization          (b) Local generalization

Figure 3: Empirical evidences of two kinds of generalization of PeVFA. (a) *Global generalization*: PeVFA shows comparable value estimation performance on the testing policy set (red) after learning on the training policy set (blue) while maintains reasonable order of optimality. (b) *Local generalization*: PeVFA ($\mathbb{V}_\theta(\chi_\pi)$) shows lower losses (i.e., closer distance to approximation target) than convention VFA ($V_\phi^\pi$) before and after the training for values of successive policies along policy improvement path, which demonstrates Corollary 2. The left axis is for approximation loss (lower is better) and the right axis is for average return as a reference of the learning process of PPO policy.

We then demonstrate local generalization of PeVFA to examine the existence of Corollary 2. We also use a 2-layer 8-unit policy network trained by PPO (Schulman et al., 2017) algorithm in OpenAI MuJoCo continuous control tasks. Parallel to the conventional value network $V_\phi^\pi(s)$ (i.e., VFA) in PPO, we set a PeVFA network $\mathbb{V}_\theta(s, \chi_\pi)$ with RPR as input. Compared with $V_\phi^\pi$, the PeVFA network $\mathbb{V}_\theta(s, \chi_\pi)$ is additionally trained with data from historical policies ($\{\pi_i\}_{i=0}^t$) along the policy improvement path. Note that $\mathbb{V}_\theta(\chi_\pi)$ does not interfere with PPO training here, and is only referred as a comparison with $V_\phi^\pi$ on the approximation error to the true values (i.e., collected returns) of successive policy $\pi_{t+1}$. To see whether the local generalization exists and can be beneficial or not, as in Figure 3(b), we illustrate the value approximation losses for $V_\phi^\pi(s)$ (red curves) and $\mathbb{V}_\theta(s, \chi_\pi)$ (blue curves), before (solid) and after (dashed) updating with on-policy samples, at each iteration during the learning process of PPO (green curves). The results demonstrate the existence and benefits of local generalization of PeVFA in InvertedPendulum-v1 and Ant-v1. By comparing approximation losses before updating (red and blue solid curves), we can observe that the good generalization of value approximation for successive policies is obtained with PeVFA $\mathbb{V}_\theta(s, \chi_\pi)$. The approximation loss of $\mathbb{V}_\theta(s, \chi_\pi)$ is almost consistently lower than that of $V_\phi^\pi(s)$ before update, providing the evidence of Corollary 2. For the dashed curves, it indicates that PeVFA can achieve lower approximation loss than conventional VFA after updating with the same on-policy samples and number of training. We emphasize the importance of the empirical evidence of beneficial local generalization, based on which we assume that PeVFA can improve conventional GPI and then propose our approaches in the following. See Appendix B.2 for complete results in 7 MuJoCo tasks and more experimental details.

### 3.3    REINFORCEMENT LEARNING WITH PEVFA

Since PeVFA extends the capacity of conventional VFA and induces value generalization among policies, we expect to derive improved RL algorithms with PeVFA by utilizing the two kinds of generalization we discussed in previous sections. In this paper, we focus on local generalization and propose a new GPI form with PeVFA. As illustrated in the right of Figure 1, the key difference is that the generalized value approximation $\mathbb{V}_{\theta_t}(\chi_{\pi_{t+1}})$ of successive policy $\pi_{t+1}$ is taken as the start point of policy evaluation at each iteration. One can see the difference by comparing $\mathbb{V}_{\theta_t}(\chi_{\pi_{t+1}})$ with the conventional one $V_{\phi_t}^{\pi_t}$, which is equivalent to $\mathbb{V}_{\theta_t}(\chi_{\pi_t})$ in definition. From Corollary 2 and the evidence in Figure 3(b), such generalized starting points are closer to approximation target (i.e., true values $v^\pi$) in some cases, thus we suggest that these local generalization steps (gray arrows) help to reduce approximation error with finite updates and improve the efficiency of the overall learning process. See a pseudo-code of GPI with PeVFA (Algorithm 2) and more discussion in Appendix C.

For global generalization, we assume that one can obtain a high-quality evaluation of unseen policies by training a PeVFA only on known policies. This indicates an appealing potential to circumvent extra sample collection and training cost for policy evaluation in many problems, e.g., evolutionary

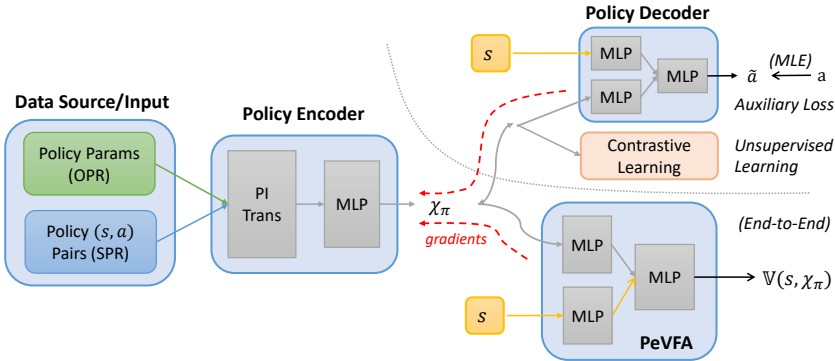

Figure 4: Framework of policy representation training. Policy network parameters (OPR) or policy state-action pairs (SPR) are fed in policy encoder with permutation-invariant (PI) transformations, producing representation $\chi_\pi$. $\chi_\pi$ can be trained by gradients from PeVFA value approximation (i.e., End-to-End), as well as with optional auxiliary action recovery loss and unsupervised contrastive learning, shown with separation by the gray dotted line.

strategy (Salimans et al., 2017), opponent modeling (He & Boyd-Graber, 2016) and policy adaptation (Arnekvist et al., 2019; Raileanu et al., 2020), which are beyond the scope of this paper.

## 4 POLICY REPRESENTATION LEARNING

In the last section, we analyze value generalization among policies induced by PeVFA and propose a new GPI for more efficient learning process. To derive practical deep RL algorithms, one key problem is policy representation, i.e., a low-dimensional embedding of RL policy. This is different from the notion of policy representation in some works (Zhou et al., 2019b; Ma et al., 2020), which is related to parameterization of policy, usually advanced policy network models or architectures. A usual policy network may have large number of parameters, thus making it inefficient and even impractical to use Raw Policy Representation (RPR) for optimization and generalization (Harb et al., 2020). To our knowledge, it remains unclear about how to obtain an effective and compact representation for an RL policy.

We propose a framework of policy representation learning as shown in Figure 4. Our intuitions come from answering the question: what makes a good policy representation, particularly when the policy is a neural network. Our proposed policy representations are motivated by a geometric perspective of the policy as a curved surface of the policy distribution; details are provided in the Appendix D.1. Concretely, we provide two policy representations below:

- *Surface Policy Representation (SPR)*: The first way is to extract policy representation from the state-action pairs (or trajectories), since they reflect the information about how policy may behave under such states. We view this as scattering sample points on policy curved surface, which should be able to capture the features as the number of samples increases.

- *Origin Policy Representation (OPR)*: Moreover, we propose a novel way to learn a low-dimensional embedding from the policy network parameters directly. Generally, we consider a policy network to be an MLP with well represented state features (e.g., features extracted by CNN for pixels or by LSTM for sequences) as input and deterministic or stochastic policy output. We then define a lossy mapping to compress all the weights and biases of the MLP to obtain the corresponding OPR.

For both SPR and OPR, we use permutation-invariant transformations, i.e., mainly use MLP then Mean-Reduce operation. The above two policy representations proposed are scalable to encode most RL policies from different stream of policy data (as illustrated in Figure 4). In this paper, we use OPR and SPR to encode the typical MLP policy. Implementation details and a discussion on encoding sophisticated RL policy like that operates images can been seen in Appendix D.2.

The related work PVN (Harb et al., 2020) proposes Network Fingerprint to circumvent the difficulty of learning policy representation from network parameters. However, it introduces additional non-

trivial optimization for a set of *probing states*. For PVFs (Faccio & Schmidhuber, 2020), all the policy weights are simply parsed as inputs to the value function without embedding, even in the nonlinear case. Besides, a few works also involve representation or embedding learning for RL policy in Multiagent Learning (Grover et al., 2018), Hierarchical RL (Wang et al., 2020), Policy Adaptation and Transfer (Hausman et al., 2018; Arnekvist et al., 2019; Raileanu et al., 2020; Harb et al., 2020). We found almost all of them belongs to the scope of SPR. A detailed review for them is provided in Appendix D.5. For OPR, our goal is to learn a representation of the policy itself from its parameters rather than of the architecture of a policy network as is common in Neural Architecture Search (NAS) (Zhou et al., 2019b; Ma et al., 2020).

The most straightforward way is to train policy representation $\chi_\pi$ together with PeVFA end-to-end as shown in Figure 4 by backpropagating the value approximation error through the policy representation. Intuitively, a good representation should be able to predict the policy's behavior. To this end, we resort to an auxiliary loss to recover the actions of $\pi$ from $\chi_\pi$ under different states (Grover et al., 2018; Raileanu et al., 2020). Concretely, an auxiliary policy decoder (or a master policy) $\bar{\pi}(\cdot|s, \chi_\pi)$ is trained through behavioral cloning, i.e., to minimize cross-entropy objective $-\sum_{(s,a)\sim\mathcal{T}_\pi} \log \bar{\pi}(a|s, \chi_\pi)$. In addition, we propose to improve the representation learning of $\chi_\pi$ with unsupervised Contrastive Learning (Oord et al., 2018; Srinivas et al., 2020; Schwarzer et al., 2020). In this way, policies are encouraged to be close to similar ones (i.e., positive samples) and to be apart from different ones (i.e., negative samples) in policy representation space. For each policy, we construct positive samples by data augmentation on policy data, depending on SPR or OPR considered; and different policies along the policy improvement path naturally provide negative samples for each other. The policy representation network is then trained with InfoNCE loss (Oord et al., 2018) as commonly done in (He et al., 2020; Chen et al., 2020). More implementation details are in Appendix D.

## 5 EXPERIMENTS

We conduct several experiments with our proposed methods and try to answer the following questions: *i.* How much does the local generalization property of PeVFA benefit a deep RL learning algorithm? *ii.* Is our policy representation learning framework effective to learn good representation for a policy network? To answer the above questions, we derive a practical deep RL algorithm with PeVFA based on Proximal Policy Optimization (PPO) (Schulman et al., 2017) to demonstrate and evaluate GPI with PeVFA and our approaches of policy representation learning.

**Setting.** Due to the popularity and simplicity in implementation, we use PPO as the base algorithm for a representative implementation of GPI with PeVFA. PPO is a policy optimization algorithm that learns both a policy network $\pi$ and a value network $V_\phi$ that approximates the values of $\pi$, typically following GPI in the left of Figure 1. The policy $\pi$ is optimized with respect to a surrogate objective (Schulman et al., 2015) using advantages calculated by $V_\phi$ and GAE (Schulman et al., 2016). We propose to replace the conventional value network $V_\phi(s)$ by a PeVFA network $\mathbb{V}_\theta(s, \chi_\pi)$ with parameters $\theta$ as given in Figure 4, and perform GPI with PeVFA as in the right of Figure 1. Without modifying the policy optimization process of vanilla PPO, we then obtain a new algorithm, called PPO-PeVFA. Note that what is actually changed here is simply a drop-and-replacement of the value function. The policy update scheme is the same for PPO and PPO-PeVFA, which means that the policy of PPO-PeVFA is not updated by gradients through policy representation feeded in $\mathbb{V}_\theta(s, \chi_\pi)$. For all the experiments in this section, we use a normal-scale policy network with 2 layers and 64 units for each layer, rather than the small policy networks used in Sec. 3.2 for demonstration. As a result, the number of policy network parameters can be over 10k for example in Ant-v1, thus prohibiting the rationality of directly using RPR for PeVFA $\mathbb{V}_\theta(s, \chi_\pi)$.

**Training.** The only difference of PPO-PeVFA is the training of policy representation $\chi_\pi$ and PeVFA network $\mathbb{V}_\theta(s, \chi_\pi)$. For policy representation $\chi_\pi$, it depends on policy data used and training losses adopted as introduced in Sec. 4. See Algorithm 3 in Appendix D.4 for an overall pseudo-code of policy representation training. For PeVFA network $\mathbb{V}_\theta(s, \chi_\pi)$, it is trained to approximate the values of multiple policies along the policy improvement (i.e. optimization) path, as Algorithm 2 in Appendix C. Both of above training use historical policies, whose data (i.e., network parameters and state-action pairs) are stored. Note that for PPO (and PPO-PeVFA), new policies are only produced after policy optimization process. We do not assume access to pre-collected policies and thus the size of policy set increases from 1 during the learning process. In our experiments, about 1k to 2k

Table 1: Max Average Return over 10 trials of 2M time steps (4M for Ant-v1). First two maximum values for each task are bolded. $\pm$ corresponds to half a std over trials.

| | Benchmarks | | Origin Policy Representation (Ours) | | | Surface Policy Representation (Ours) | | |
|---|---|---|---|---|---|---|---|---|
| Environments | PPO | Ran PR | E2E | CL | AUX | E2E | CL | AUX |
| HalfCheetah-v1 | $2621 \pm 259.31$ | $2470 \pm 291.65$ | $3171 \pm 427.63$ | $\mathbf{3725 \pm 348.55}$ | $3175 \pm 517.52$ | $2774 \pm 233.39$ | $\mathbf{3349 \pm 341.42}$ | $3216 \pm 506.39$ |
| Hopper-v1 | $1639 \pm 294.47$ | $1226 \pm 348.10$ | $2085 \pm 310.91$ | $2351 \pm 231.11$ | $2214 \pm 360.78$ | $2227 \pm 297.35$ | $\mathbf{2392 \pm 263.93}$ | $\mathbf{2577 \pm 217.73}$ |
| Walker2d-v1 | $1505 \pm 320.55$ | $1269 \pm 209.61$ | $1856 \pm 305.51$ | $2038 \pm 315.51$ | $\mathbf{2044 \pm 316.32}$ | $1930.57 \pm 456.02$ | $\mathbf{2203 \pm 381.95}$ | $1980 \pm 325.54$ |
| Ant-v1 | $2835 \pm 152.04$ | $2742 \pm 71.11$ | $3581 \pm 185.43$ | $\mathbf{4019 \pm 162.47}$ | $\mathbf{3784 \pm 268.99}$ | $3173 \pm 184.75$ | $3632 \pm 134.27$ | $3397 \pm 200.03$ |
| InvDouPend-v1 | $9344 \pm 11.02$ | $9355 \pm 0.40$ | $\mathbf{9357 \pm 0.29}$ | $9355 \pm 0.64$ | $9355 \pm 0.68$ | $9355 \pm 0.89$ | $\mathbf{9356 \pm 0.96}$ | $9355 \pm 1.42$ |
| LunarLander-v2 | $219 \pm 5.33$ | $226 \pm 2.83$ | $\mathbf{238 \pm 3.37}$ | $\mathbf{239 \pm 3.70}$ | $234 \pm 3.47$ | $236 \pm 3.13$ | $234 \pm 3.13$ | $235 \pm 5.70$ |

policies are produced in an experimental trial. We use all historical policies in our training and for the case with many more policies, other sophisticated training methods can be considered.

**Results.** We conduct the following experiments in OpenAI Gym MuJoCo continuous control tasks (Brockman et al., 2016; Todorov et al., 2012). Beside the comparison to PPO, we consider another benchmark, PPO-PeVFA with fixed randomly generated policy representation for each policy, namely Ran PR. The overall experimental results are in Table 1. More experimental details are in Appendix F.1 and complete learning curves are provided in Appendix F.2 due to space limitation.

**Question 1.** *Can PPO-PeVFA with only End-to-End (E2E) policy representation outperform PPO?*

From Table 1, we can find that both PPO-PeVFA with OPR and with SPR trained in E2E fashion outperforms PPO in all 6 tasks, especially in Ant-v1. This indicates that effective policy representation can be learned from value approximation loss in our experiments, based on which PeVFA can benefit from local generalization thus improve the learning process. Additionally, Ran PR shows an overall degeneration compared to PPO, demonstrating the significance of OPR and SPR.

**Question 2.** *Can representation learned through contrastive learning (CL) and auxiliary loss (AUX) further benefit performance of PPO-PeVFA?*

For both OPR and SPR, we observe consistent improvements induced by CL and AUX over E2E, which means that additional learning to maintain unsupervised consistency (CL) and capture policy behavior (AUX) further benefits policy representation, and eventually results in substantial improvement over PPO in all tasks except for the first two simple tasks. In an overall view, OPR slightly outperforms SPR and as CL does over AUX. We hypothesize that it is due to the stochasticity of state-action pairs which serve as inputs of SPR and training samples for AUX. We suggest that this remains space for future improvement.

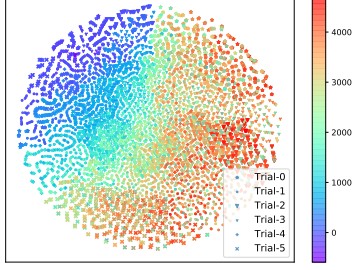

Figure 5: t-SNE visualization for OPR (E2E) of 6k policies from 5 trials (denoted by different markers) in Ant-v1. Policies are colored by average return.

**Question 3.** *How is the learned representation like when viewing in a low-dimensional space?*

We show a 2D t-SNE (Maaten & Hinton, 2008) view in Figure 5 and more visualisations in Appendix F. Policies from different trials are locally continuous (multimodality) while a global evolvement emerges with respect to policy performance.

## 6 CONCLUSION AND FUTURE WORK

In this paper, we propose Policy-extended Value Function Approximator (PeVFA) which induces value generalization among policies. We propose a new form of GPI with PeVFA which can potentially benefit from local value generalization along the policy improvement path. Moreover, we propose several approaches to learn an effective low-dimensional representation of an RL policy. Our experiments demonstrate the effectiveness of the generalization properties of PeVFA and our proposed methods of representing the RL policy.

Our work opens up new ways to potentially improve the policy learning process from global or local perspectives using PeVFAs that receive an explicit policy representation to improve generalization of the value approximation. We believe our work can guide future research directions on policy

representations and PeVFA in many RL problems. We plan to further study policy representation learning, as well as more complex value generalization scenarios, including TD learning with non-stationary approximation targets. In addition, optimizing policies in the representation space can also be an interesting direction.

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

# A PROOFS

## A.1 PROOF OF THEOREM 1

*Proof.* Start from Assumption 2, we derive the upper bound of $\hat{f}(\pi_2)$:

$$\hat{f}(\pi_2) \leq \hat{f}(\pi_1) + \mathcal{M}(\pi_1, \pi_2, \hat{L}), \tag{3}$$

where $\mathcal{M}(\pi_1, \pi_2, \hat{L})$ can have different forms that depends on the specific smoothness property, for examples:

$$\mathcal{M}(\pi_1, \pi_2, \hat{L}) = \begin{cases} \hat{L}_0 \cdot d(\pi_1, \pi_2) & \text{①} \\ \langle \hat{f}'(\pi_1), \pi_2 - \pi_1 \rangle + \frac{1}{2}\hat{L}_1 \cdot d(\pi_1, \pi_2)^2 & \text{②} \\ \langle \hat{f}'(\pi_1), \pi_2 - \pi_1 \rangle + \frac{1}{2}\langle \hat{f}''(\pi_1)(\pi_2 - \pi_1), \pi_2 - \pi_1 \rangle + \frac{1}{6}\hat{L}_2 \cdot d(\pi_1, \pi_2)^3 & \text{③} \end{cases}$$

①, ②, ③ correspond to Lipschitz Continuous, Lipschitz Gradients and Lipschitz Hessian (Nesterov & Polyak, 2006).

Combined with Assumption 1 and consider the case $f(\pi_1) \leq f(\pi_2)$, Equation 3 can be further transformed as follows:

$$\begin{aligned} \hat{f}(\pi_2) &\leq \hat{f}(\pi_1) + \mathcal{M}(\pi_1, \pi_2, \hat{L}) \\ &\leq \gamma f(\pi_1) + \mathcal{M}(\pi_1, \pi_2, \hat{L}) \\ &\leq \underbrace{\gamma f(\pi_2)}_{\text{generalized contraction}} + \underbrace{\mathcal{M}(\pi_1, \pi_2, \hat{L})}_{\text{locality margin}}, \end{aligned} \tag{4}$$

which yields the generalization upper bound in Theorem 1. We note the first term of RHS of Equation 4 as *generalized contraction* since it is from the contraction on $f(\pi_1)$, and the second term as *locality margin* since it is determined by specific local property. □

## A.2 PROOF OF COROLLARY 1

*Proof.* Consider the concrete form of locality margin $\mathcal{M}(\pi_1, \pi_2, \hat{L})$ in the case of Lipschitz continuous, i.e.,

$$\hat{f}(\pi_2) \leq \gamma f(\pi_2) + \hat{L}_0 \cdot d(\pi_1, \pi_2), \tag{5}$$

and consider $f(\pi_2) \neq 0$ (note that $f(\pi_2) \geq 0$):

$$\hat{f}(\pi_2) \leq \gamma_g f(\pi_2), \text{ where } \gamma_g = \gamma + \frac{\hat{L}_0 \cdot d(\pi_1, \pi_2)}{f(\pi_2)}. \tag{6}$$

Then we get the contraction condition of value generalization on $\pi_2$ in Corollary 1, by letting the RHS of Equation 5 not greater than $f(\pi_2)$, i.e., $\gamma_g \leq 1$ (note that $\gamma \in (0, 1]$):

$$\begin{aligned} \gamma f(\pi_2) + \hat{L}_0 \cdot d(\pi_1, \pi_2) &\leq f(\pi_2) \\ (1 - \gamma)f(\pi_2) &\geq \hat{L}_0 \cdot d(\pi_1, \pi_2) \\ f(\pi_2) &\geq \frac{\hat{L}_0 \cdot d(\pi_1, \pi_2)}{1 - \gamma}. \end{aligned} \tag{7}$$

Intuitions observed from Equation 7 are discussed in Remark 3. In another word, the tighter the contraction on learned policy $\pi_1$ is, the closer the two policies are, the smoother the approximation loss function $\hat{f}$ is, the generalization on unlearned policy $\pi_2$ is better. □

Corollary 1 provides the generalization contraction condition on $f(\pi_2)$. We also provide another view from the perspective of $f(\pi_1)$, by additionally considering the smoothness property of $f$.

Similar as in Assumption 2, we assume that $f$ is Lipschitz continuous at $\pi_1$ with Lipschitz constant $L_0$, e.g., $|f(\pi) - f(\pi_1)| \leq L_0 \cdot d(\pi, \pi_1)$ with some metric space $(\Pi, d)$. Higher-order smoothness property can also be considered here. We then obtain a lower bound of $f(\pi_2)$ below:

$$f(\pi_1) - L_0 \cdot d(\pi, \pi_1) \leq f(\pi_2). \tag{8}$$

We then chain the second line of RHS in Equation 4 ($\mathcal{M}$ in the case of Lipschitz continuous) and the LHS of Equation 8:

$$\hat{f}(\pi_2) \leq \gamma f(\pi_1) + \hat{L}_0 \cdot d(\pi, \pi_1) \leq f(\pi_1) - L_0 \cdot d(\pi, \pi_1) \leq f(\pi_2), \tag{9}$$

re-arrange and yield the generalization contraction condition on $f(\pi_1)$:

$$\begin{aligned}
\gamma f(\pi_1) + \hat{L}_0 \cdot d(\pi 1, \pi_2) &\leq f(\pi_1) - L_0 \cdot d(\pi_1, \pi_2) \\
(1 - \gamma)f(\pi_1) &\geq (L_0 + \hat{L}_0)d(\pi_1, \pi_2) \\
f(\pi_1) &\geq \frac{(L_0 + \hat{L}_0)d(\pi_1, \pi_2)}{1 - \gamma}.
\end{aligned} \tag{10}$$

### A.3 Proof of Corollary 2

*Proof.* Due to *Triangle Inequality*, we have:

$$\begin{aligned}
f_{\theta_t}(\pi_t) + \|\mathbb{V}_{\theta_t}(\pi_t) - \mathbb{V}_{\theta^*}(\pi_{t+1})\| &\geq \|\mathbb{V}_{\theta^*}(\pi_t) - \mathbb{V}_{\theta^*}(\pi_{t+1})\| \\
\text{i.e., } \|\mathbb{V}_{\theta_t}(\pi_t) - \mathbb{V}_{\theta^*}(\pi_t)\| + \|\mathbb{V}_{\theta_t}(\pi_t) - \mathbb{V}_{\theta^*}(\pi_{t+1})\| &\geq \|\mathbb{V}_{\theta^*}(\pi_t) - \mathbb{V}_{\theta^*}(\pi_{t+1})\|
\end{aligned} \tag{11}$$

Combined with condition

$$\begin{aligned}
f_{\theta_t}(\pi_t) + f_{\theta_t}(\pi_{t+1}) &\leq \|\mathbb{V}_{\theta^*}(\pi_t) - \mathbb{V}_{\theta^*}(\pi_{t+1})\| \\
\text{i.e., } \|\mathbb{V}_{\theta_t}(\pi_t) - \mathbb{V}_{\theta^*}(\pi_t)\| + \|\mathbb{V}_{\theta_t}(\pi_{t+1}) - \mathbb{V}_{\theta^*}(\pi_{t+1})\| &\leq \|\mathbb{V}_{\theta^*}(\pi_t) - \mathbb{V}_{\theta^*}(\pi_{t+1})\|.
\end{aligned} \tag{12}$$

Chain above two inequality,

$$\begin{aligned}
f_{\theta_t}(\pi_t) + f_{\theta_t}(\pi_{t+1}) \leq \|\mathbb{V}_{\theta^*}(\pi_t) - \mathbb{V}_{\theta^*}(\pi_{t+1})\| &\leq f_{\theta_t}(\pi_t) + \|\mathbb{V}_{\theta_t}(\pi_t) - \mathbb{V}_{\theta^*}(\pi_{t+1})\| \\
\underbrace{f_{\theta_t}(\pi_{t+1})}_{\text{generalizated VAD with PeVFA}} &\leq \underbrace{\|\mathbb{V}_{\theta_t}(\pi_t) - \mathbb{V}_{\theta^*}(\pi_{t+1})\|}_{\text{conventional VAD}},
\end{aligned} \tag{13}$$

then we have that with local generalization of values for successive policy $\pi_{t+1}$, the value approximation distance (VAD) can be closer in contrast to the conventional one (RHS of Equation 13). $\square$

In practice, it is also possible for farther distance to exist, e.g., the condition in above Corollary is not satisfied. Moreover, under nonlinear function approximation, it is not necessary that a closer approximation distance (induced by Corollary 2) ensures easier approximation or optimization process. This is related to many things, e.g., the underlying function space, the optimization landscape, the learning algorithm used and etc. In this paper, we provide a condition for potentially beneficial local generalization and we resort to empirical examination as shown in Sec. 3.2. Further study of beneficial cases especially under nonlinear function approximation is planned for future work.

## B Details of Empirical Evidence of Two Kinds of Generalization

### B.1 Global Generalization in 2D Point Walker

Global generalization denotes the generalization scenario that values can generalize to unlearned policies ($\pi' \in \Pi_1$) from already learned policies ($\pi \in \Pi_0$). We conduct the following experiments to demonstrate global generalization in a 2D continuous Point Walker environment with synthetic simple policies.

**Environment.** We consider a point walker on a 2D continuous plane with

- state: $(x, y, \sin(\theta), \cos(\theta), \cos(x), \cos(y))$, where $\theta$ is the angle of the polar coordinates,
- action: 2D displacement, $a \in \mathbb{R}^2_{[-1,1]}$,
- a deterministic transition function that describes the locomotion of the point walker, depending on the current position and displacement issued by agent, i.e., $\langle x', y' \rangle = \langle x, y \rangle + a$,
- a reward function: $r_t = \frac{u_{t+1} - u_t}{10}$ with utility $u_t = x_t^2 - y_t^2$, as illustrated in Figure 6(a).

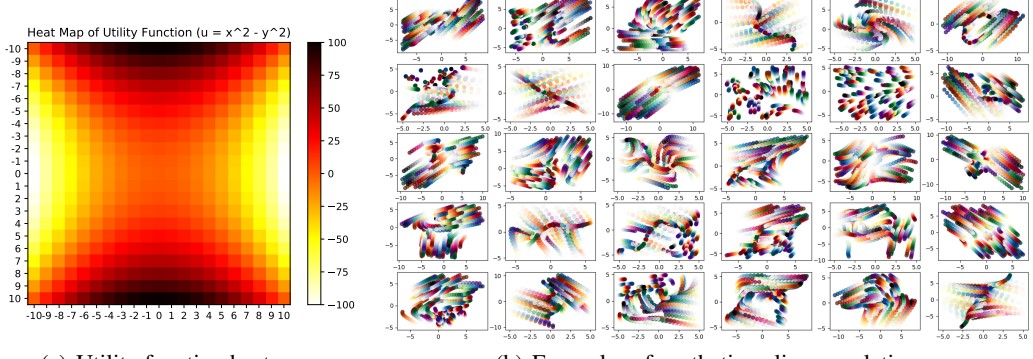

(a) Utility function heat map
(b) Examples of synthetic policy population

Figure 6: 2D Pointer Walker. (a) The heat map of the utility function of the 2D plane. The darker regions have higher utilities. (b) Demonstrative illustrations of trajectories generated by 30 synthetic policies, showing diverse behaviors and patterns. Each subplot illustrates the trajectories generated in 50 episodes by a randomly synthetic policy, with different colors as separation. For each trajectory (the same color in one subplot), transparency represents the dynamics along timesteps, i.e., fully transparent and non-transparent denotes the positions at first and last timesteps.

**Synthetic Policy.** We build the policy sets $\Pi = \Pi_0 \cup \Pi_1$ and $\Pi_0 \cap \Pi_1 = \emptyset$ with synthetic policies. Each synthetic policy is a 2-layer *tanh*-activated neural policy network with random initialization. Each policy is deterministic, taking an environmental state as input and outputting a displacement in the plane. We find that the synthetic population generated by such a simple way can show diverse behaviors. Figure 6(b) shows the motion patterns of an example of such a synthetic population. Note that the synthetic policies are not trained in this experiment.

**Policy Dataset.** We rollout each policy in environment to collect trajectories $\mathcal{T} = \{\tau_i\}_{i=0}^k$. For such small synthetic policies, it is convenient to obtain policy representation. Here we use the concatenation of all weights and biases of the policy network (22 in total) as representation $\chi_\pi$ for each policy $\pi$, called *raw* policy representation (RPR). Therefore, combined with the trajectories collected, we obtain the policy dataset, i.e., $\{(\chi_{\pi_j}, \mathcal{T}_{\pi_j})\}_{j=0}^n$. In total, 20k policies are synthesized in our experiments and we collected 50 trajectories with horizon 10 for each policy.

We separate the synthetic policies into training set (i.e., unknown policies $\Pi_0$) and testing set (i.e., unseen policies $\Pi_1$) in a proportion of $8 : 2$. We set a PeVFA network $\mathbb{V}_\theta(s, \chi_\pi)$ to approximate the values of training policies (i.e., $\pi \in \Pi_0$), and then conduct evaluation on testing policies (i.e., $\pi \in \Pi_1$). We use Monte Carlo return (Sutton & Barto, 1998) of collected trajectories as approximation target (true value of policies) in this experiment. The network architecture of $\mathbb{V}_\theta(s, \chi_\pi)$ is illustrated in Figure 7. The learning rate is 0.005, batch size is 256. K-fold validation is performed through shuffling training and testing sets.

Figure 2(a) shows that a PeVFA trained with data collected by training set $\Pi_0$ achieves reasonable value prediction of unseen testing policies in $\Pi_1$, as well as maintains the order of optimality among testing policies. This indicates that value generalization can exist among policy space with a properly trained PeVFA. RPR can also be one alternative of policy representation when policy network is of small scale.

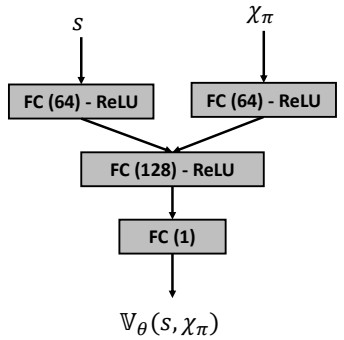

Figure 7: An illustration of architecture of PeVFA network. FC is abbreviation for Fully-connected layer.

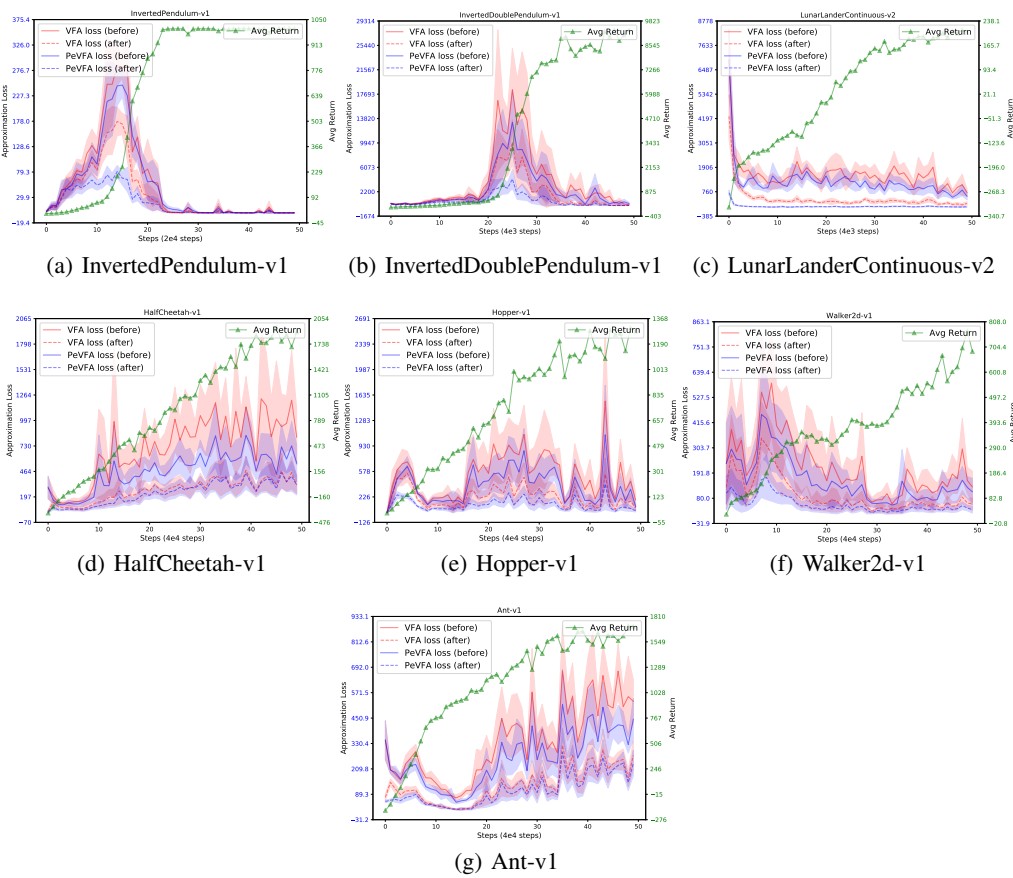

Figure 8: Complete empirical evidence of local generalization of PeVFA across 7 MuJoCo tasks. The learning rate of policy and value function approximators are 0.0001 and 0.001 respectively. Each plot has two vertical axes, the left one for approximation error (red and blue curves) and the right one for average return (green curves). Red and blue denotes the approximation error of conventional VFA ($V_\phi^\pi(s)$) and of PeVFA ($\mathbb{V}_\theta(s, \chi_\pi)$) respectively; solid and dashed curves denote the approximation error before and after the training for values of successive policy (i.e., policy evaluation) with conventional VFA and PeVFA, averaged over 6 trials. The shaded region denotes half a standard deviation of average evaluation. PeVFA consistently shows lower losses (i.e., closer to approximation target) across all tasks than convention VFA before and after policy evaluation along policy improvement path, which demonstrates Corollary 2.

## B.2 LOCAL GENERALIZATION IN MUJOCO CONTINUOUS CONTROL TASKS

We demonstrate local generalization of PeVFA, especially to examine the existence of Corollary 2, i.e., PeVFA can induce closer approximation distance (i.e., lower approximation error) than conventional VFA along the policy improvement path.

We use a 2-layer 8-unit policy network trained by PPO (Schulman et al., 2017) algorithm in OpenAI MuJoCo continuous control tasks. As in previous section, using a very small policy network is for the convenience of training and acquisition of policy representation in this demonstrative experiment. We use all weights and biases of the small policy network (also called *raw* policy representation, RPR), whose number is about 10 to 100 in our experiments, depending on the specific environment (i.e., the state and action dimensions). We train the small policy network as commonly done with PPO (Schulman et al., 2017) and GAE (Schulman et al., 2016). The convention value network $V_\phi^\pi(s)$ (VFA), is a 2-layer 128-unit ReLU-activated MLP with state as input and value as output. Parallel to the conventional VFA in PPO, we set a PeVFA network $\mathbb{V}_\theta(s, \chi_\pi)$ with RPR as

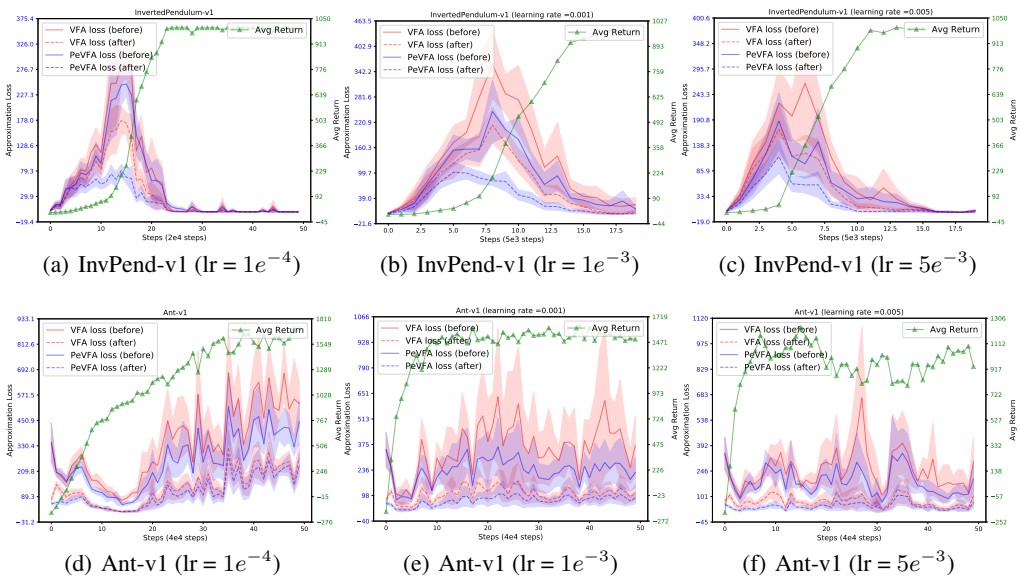

Figure 9: Empirical evidence of local generalization of PeVFA on InvertedPendulum-v1 and Ant-v1 with different learning rates of policy, i.e., $\{0.0001, 0.001, 0.005\}$. Results are averaged over 6 trials.

additional input. The structure of PeVFA differs at the first hidden layer which has two input streams and each of them has 64 units, as illustrated in Figure 7, so that making VFA and PeVFA have similar scales of parameter number. In contrast to convention VFA $V_\phi^\pi$ which approximates the value of current policy (e.g., Algorithm 1), PeVFA $\mathbb{V}_\theta(s, \chi_\pi)$ has the capacity to preserve values of multiple policies and thus is additionally trained to approximate the values of all historical policies ($\{\pi_i\}_{i=0}^t$) along the policy improvement path (e.g., Algorithm 2). The learning rate of policy is 0.0001 and the learning rate of value function approximators ($V_\phi^\pi(s)$ and $\mathbb{V}_\theta(s, \chi_\pi)$) is 0.001. The training scheme of PPO policy here is the same as that described in Appendix F.1 and Table 2.

Note that $\mathbb{V}_\theta(\chi_\pi)$ does not interfere with PPO training here, and is only referred as a comparison with $V_\phi^\pi$ on the approximation error to the true values of successive policy $\pi_{t+1}$. We use the MC returns of on-policy data (i.e., trajectories) collected by current successive policy as unbiased estimates of true values, similarly done in (v. Hasselt, 2010; Fujimoto et al., 2018). Then we calculate the approximation error for VFA $V_\phi^\pi$ and PeVFA $\mathbb{V}_\theta(\chi_\pi)$ to the approximation target before and after value network training of current iteration. Finally, we compare the approximation error between VFA and PeVFA to approximately examine local generalization and closer approximation target in Corollary 2. Complete results of local generalization across all 7 MuJoCo tasks are show in Figure 8. The results show that PeVFA consistently shows lower losses (i.e., closer to approximation target) across all tasks than convention VFA before and after policy evaluation along policy improvement path, which demonstrates Corollary 2. Moreover, we also provide similar empirical evidence when policy is updated with larger learning rates, as in Figure B.1.

A common observation across almost all results in Figure 8 and in Figure B.1 is that the larger the extent of policy change (see the regions with a sheer slope on green curves), the higher the losses of conventional VFA tend to be (see the peaks of red curves), where the generalization tends to be better and more significant (see the blue curves). Since InvertedPendulum-v1 is a simple task while the complexity of the solution for Ant-v1 is higher, the difference between value approximation losses of PeVFA and VFA is more significant at the regions with fast policy improvement. Besides, the Raw Policy Representation (RPR) we used here does not necessarily induce a smooth and efficient policy representation space, among which policy values are easy to generalize and optimize. Thus, RPR may be sufficient for a good generalization in InvertedPendulum-v1 but may be not in Ant-v1. Overall, we think that the quantity of value approximation loss is related to several factors of the environment such as the reward scale, the extent of policy change, the complexity of underlying so-

lution (e.g., value function space) and some others. A further investigation on this can be interesting.

## C  GENERALIZED POLICY ITERATION WITH PEVFA

### C.1  COMPARISON BETWEEN CONVENTIONAL GPI AND GPI WITH PEVFA

A graphical comparison of conventional GPI and GPI with PeVFA is shown in Figure 1. Here we provide another comparison with pseudo-codes.

From the lens of Generalized Policy Iteration (Sutton & Barto, 1998), for most model-free policy-based RL algorithms, the approximation of value function and the update of policy through policy gradient theorem are usually conducted iteratively. Representative examples are REINFORCE (Sutton & Barto, 1998), Advantage Actor-Critic (Mnih et al., 2016), Deterministic Policy Gradient (DPG) (Silver et al., 2014) and Proximal Policy Optimization (PPO) (Schulman et al., 2017). With conventional value function (approximator), policy evaluation is usually performed in an on-policy or off-policy fashion. We provide a general GPI description of model-free policy-based RL algorithm with conventional value functions in Algorithm 1.

---

**Algorithm 1** Generalized policy iteration for model-free policy-based RL algorithm with conventional value functions ($V^\pi(s)$ or $Q^\pi(s, a)$)

1: Initialize policy $\pi_0$ and $V^\pi_{-1}(s)$ or $Q^\pi_{-1}(s, a)$
2: Initialize experience buffer $\mathcal{D}$
3: **for** iteration t $= 0, 1, 2, \dots$ **do**
4:     Rollout policy $\pi_t$ in the environment and obtain trajectories $\mathcal{T}_t = \{\tau_i\}_{i=0}^k$
5:     Add experiences $\mathcal{T}_t$ in buffer $\mathcal{D}$
6:     **if** *on-policy update* **then**
7:         Prepare training samples from rollout trajectories $\mathcal{T}_t$
8:     **else if** *off-policy update* **then**
9:         Prepare training samples by sampling from buffer $\mathcal{D}$
10:     **end if**
11:     Calculate approximation target $\{y_i\}_i$ from training samples (e.g., with MC or TD)
12:     # Generalized Policy Evaluation
13:     Update $V^\pi_{t-1}(s)$ or $Q^\pi_{t-1}(s, a)$ with $\{(s_i, y_i)\}_i$ or $\{(s_i, a_i, y_i)\}_i$, i.e., $V^\pi_t \longleftarrow V^\pi_{t-1}$ or $Q^\pi_t \longleftarrow Q^\pi_{t-1}$
14:     # Generalized Policy Improvement
15:     Update policy $\pi_t$ with regard to $V^\pi_t(s)$ or $Q^\pi_t(s, a)$ through policy gradient theorem, i.e., $\pi_{t+1} \longleftarrow \pi_t$
16: **end for**

---

Note that we use subscript $t - 1 \rightarrow t$ (Line 13 in Algorithm 1) to let the updated value functions to correspond to the evaluated policy $\pi_t$ during policy evaluation process in current iteration.

As a comparison, a new form of GPI with PeVFA is shown in Algorithm 2. Except for the different parameterization of value function, PeVFA can perform additionally training on historical policy experiences at each iteration (Line 7-8). This is naturally compatible with PeVFA since it develops the capacity of conventional value function to preserve the values of multiple policies. Such a training is to improve the value generalization of PeVFA among a policy set or policy space. Note that for value approximation of current policy $\pi_t$ (Line 10-14), the start points are generalized values of $\pi_t$ from historical approximation, i.e., $\mathbb{V}_{t-1}(s, \chi_{\pi_t})$ and $\mathbb{Q}_{t-1}(s, a, \chi_{\pi_t})$. In another word, this is the place where local generalization steps (illustrated in Figure 2(b)) are. One may compare with conventional start points ($V^\pi_{t-1}(s)$ and $Q^\pi_{t-1}(s, a)$, Line 13 in Algorithm 1) and see the difference, e.g., $V^\pi_{t-1}(s) \Leftrightarrow V^{\pi_{t-1}}(s) \Leftrightarrow \mathbb{V}_{t-1}(s, \chi_{\pi_{t-1}})$ is different with $\mathbb{V}_{t-1}(s, \chi_{\pi_t})$, where $\Leftrightarrow$ is used to denote an equivalence in definition. As discussed in Sec. 3.2 and 3.3, we suggest that such local generalization steps help to reduce approximation error and thus improve efficiency during the learning process.

---

**Algorithm 2** Generalized policy iteration of model-free policy-based RL algorithm with PeVFAs ($\mathbb{V}(s, \chi_\pi)$ or $\mathbb{Q}(s, a, \chi_\pi)$)

---

1: Initialize policy $\pi_0$ and PeVFA $\mathbb{V}_{-1}(s, \chi_\pi)$ or $\mathbb{Q}_{-1}(s, a, \chi_\pi)$
2: Initialize experience buffer $\mathcal{D}$
3: **for** iteration t $= 0, 1, 2, \ldots$ **do**
4:     Rollout policy $\pi_t$ in the environment and obtain trajectories $\mathcal{T}_t = \{\tau_i\}_{i=0}^k$
5:     Get the policy representation $\chi_{\pi_t}$ for policy $\pi_t$ (from policy network parameters or policy rollout experiences)
6:     Add experiences $(\chi_{\pi_t}, \mathcal{T}_t)$ in buffer $\mathcal{D}$
7:     # Value approximation training for historical policies $\{\pi_i\}_{i=0}^{t-1}$
8:     Update PeVFA $\mathbb{V}_{t-1}(s, \chi_{\pi_i})$ or $\mathbb{Q}_{t-1}(s, a, \chi_{\pi_i})$ with all historical policy experiences $\{(\chi_{\pi_i}, \mathcal{T}_i)\}_{i=0}^{t-1}$
9:     # Conventional value approximation training for current policy $\pi_t$
10:     **if** *on-policy update* **then**
11:         Update PeVFA $\mathbb{V}_{t-1}(s, \chi_{\pi_t})$ or $\mathbb{Q}_{t-1}(s, a, \chi_{\pi_t})$ for $\pi_t$ with on-policy experiences $(\chi_{\pi_t}, \mathcal{T}_t)$
12:     **else if** *off-policy update* **then**
13:         Update PeVFA $\mathbb{V}_{t-1}(s, \chi_{\pi_t})$ or $\mathbb{Q}_{t-1}(s, a, \chi_{\pi_t})$ for $\pi_t$ with off-policy experiences $\chi_{\pi_t}$ and $\{\mathcal{T}_i\}_{i=0}^t$ from experience buffer $\mathcal{D}$
14:     **end if**
15:     $\mathbb{V}_t \longleftarrow \mathbb{V}_{t-1}$ or $\mathbb{Q}_t \longleftarrow \mathbb{Q}_{t-1}$
16:     Update policy $\pi_t$ with regard to $\mathbb{V}_t(s, \chi_{\pi_t})$ or $\mathbb{Q}_t(s, a, \chi_{\pi_t})$ through policy gradient theorem, i.e., $\pi_{t+1} \longleftarrow \pi_t$
17: **end for**

---

## C.2 MORE DISCUSSIONS ON PEVFA

**Off-Policy Learning.** Off-policy Value Estimation (Sutton & Barto, 1998) denotes to evaluate the values of some target policy from data collected by some behave policy. As commonly seen in RL (also shown in Line 6-10 in Algorithm 1), different algorithms adopt on-policy or off-policy methods. For GPI with PeVFA, especially for the value estimation of historical policies (Line 8 in Algorithm 2), on-policy and off-policy methods can also be considered here. One interesting thing is, in off-policy case, one can use experiences from any policy for the learning of another one, which can be appealing since the high data efficiency of value estimation of each policy can strengthen value generalization among themselves with PeVFA, which further improve the value estimation process.

**Convergence of GPI with PeVFA.** Convergence of GPI is usually discussed in some ideal cases, e.g., with small and finite state action spaces and with sufficient function approximation ability. In this paper, we focus on the comparison between conventional VFA and PeVFA in value estimation, i.e., Policy Evaluation, and we make no assumption on the Policy Improvement part. We conjecture that with the same policy improvement algorithm and sufficient function approximation ability, GPI with conventional VFA and GPI with PeVFA finally converge to the same policy. Moreover, based on Corollary 2 and our empirical evidence in Sec. 3.2, GPI with PeVFA can be more efficient in some cases: with local generalization, it could take less experiences (training) for PeVFA to reach the same level of approximation error than conventional VFA, or with the same amount of experience (training), PeVFA could achieve lower approximation error than conventional VFA. We believe that a deeper dive in convergence analysis is worth further investigation.

**PeVFA with TD Value Estimation.** In this paper, we propose PPO-PeVFA as a representative instance of re-implementing DRL algorithms with PeVFA. Our theoretical results and algorithm 2 proposed under the general policy iteration (GPI) paradigm are suitable for TD value estimation as well in principle. One potential thing that deserves further investigation is that, it can be a more complex generalization problem since the approximation target of TD learning is moving (in contrast to the stationary target when unbiased Monte Carlo estimates are used). The non-stationarity induced by TD is recognized to hamper the generalization performance in RL as pointed out in recent work (Igl et al., 2020). Further study on PeVFA with TD learning (e.g., TD3 and SAC) is planned in the future as mentioned in Sec. 6.

# D   POLICY REPRESENTATION LEARNING DETAILS

## D.1   POLICY GEOMETRY

A policy $\pi \in \Pi = \mathcal{P}(\mathcal{A})^{\mathcal{S}}$, defines the behavior (action distribution) of the agent under each state. For a more intuitive view, we consider the geometrical shape of a policy: all state $s \in \mathcal{S}$ and all action $a \in \mathcal{A}$ are arranged along the $x$-axis and $y$-axis of a 2-dimensional plane, and the probability (density) $\pi(a|s)$ is the value of $z$-axis over the 2-dimensional plane. Note that for finite state space and finite action space (discrete action space), the policy can be viewed as a $|\mathcal{S}| \times |\mathcal{A}|$ table with each entry in it is the probability of the corresponding state-action case. Without loss of generality, we consider the continuous state and action space and the policy geometry here. Illustrations of policy geometry examples are shown in Figure 10.

Figure 10(a) shows the policy geometry in a general case, where the policy can be defined arbitrarily. Generally, the policy geometry can be any possible geometrical shape (s.t. $\forall s \in \mathcal{S}, \sum_{a \in \mathcal{A}} \pi(a|s) = 1$). This means that the policy geometry is not necessarily continuous or differentiable in a general case. Specially, one can imagine that the geometry of a deterministic policy consists of peak points ($z = 1$) for each state and other flat regions ($z = 0$). Figure 10(b) shows an example of synthetic continuous policy which can be viewed as a 3D curved surface. In Deep RL, a policy may usually be modeled as a deep neural network. Assume that the neural policy is a function that is almost continuous and differentiable everywhere, the geometry of such a neural policy can also be continuous and differentiable almost everywhere. As shown in Figure 10(c), we provide a demo of neural policy by smoothing an arbitrary policy along both state and action axes.

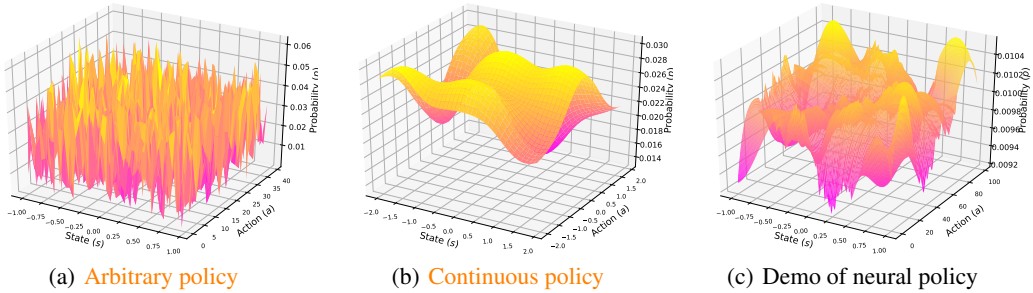

(a) Arbitrary policy       (b) Continuous policy       (c) Demo of neural policy

Figure 10:   Examples of policy geometry. (a) An arbitrary policy, where $p(s, a)$ is sampled from $\mathcal{N}(0, 1)$ for a joint space of 40 states and 40 actions and then normalized along action axis. States are squeezed into the range of $[-1, 1]$ for clarity. (b) A synthetic continuous policy with $p(s, a) = (1 - a^5 + s^5) \exp(-s^2 - a^2)$ for a joint space of $s \in [-2, 2]$ and $a \in [-2, 2]$ (each of which are discretized into 40 ones) and then normalized along action axis. (c) A general demo of neural network policy, generated from an arbitrary policy (as in (a)) over a joint space of 200 states and 100 actions with some smoothing skill. States are squeezed into the range of $[-1, 1]$ for clarity and the probability masses of actions under each state are normalized to sum into 1.

## D.2   IMPLEMENTATION DETAILS OF SURFACE POLICY REPRESENTATION (SPR) AND ORIGIN POLICY REPRESENTATION (OPR)

Here we provide a detailed description of how to encode different policy data for Surface Policy Representation (SPR) and Origin Policy Representation (OPR) we introduced in Sec. 4.

**Encoding of State-action Pairs for SPR.** Given a set of state-action pairs $\{s_i, a_i\}_{i=1}^n$ (with size $[n, s\_dim + a\_dim]$) generated by policy $\pi$ (i.e., $a_i \sim \pi(\cdot|s_i)$), we concatenate each state-action pair and obtain an embedding of it by feeding it into an MLP, resulting in a stack of state-action embedding with size $[n, e\_dim]$. After this, we perform a mean-reduce operator on the stack and obtain an SPR with size $[1, e\_dim]$. A similar permutation-invariant transformation is previously adopted to encode trajectories in (Grover et al., 2018).

**Encoding of Network Parameters for OPR.** We propose a novel way to learn low-dimensional embedding from policy network parameters directly. To our knowledge, we are the first to learn

policy embedding from neural network parameters in RL. Note that latent space of neural networks are also studied in differentiable Network Architecture Search (NAS) (Liu et al., 2019; Luo et al., 2018), where architecture-level embedding are usually considered. In contrast, OPR cares about parameter-level embedding with a given architecture.

Consider a policy network to be a MLP with well-represented state (e.g., CNN for pixels, LSTM for sequences) as input and deterministic or stochastic policy output. We compress all the weights and biases of the MLP to obtain an OPR that represents the decision function. The encoding process of an MLP with two hidden layers is illustrated in Figure 11. The main idea is to perform permutation-invariant transformation for inner-layer weights and biases for each layer first. For each unit of some layer, we view the unit as a non-linear function of all outputs, determined by weights, a bias term and activation function. Thus, the whole layer can be viewed as a batch of operations of previous outputs, e.g., with the shape $[h_t, h_{t-1} + 1]$ for $t \geq 1$ and $t = 0$ is also for the input layer. Note that we neglect activation function in the encoding since we consider the policy network structure is given. That is also why we call OPR as parameter-level embedding in contrast to architecture-level enbedding in NAS (mentioned in the last paragraph). We then feed the operation batch in an MLP and perform mean-reduce to outputs. Finally we concatenate encoding of layers and obtain the OPR.

We use permutation-invariant transformation for OPR because that we suggest the operation batch introduced in the last paragraph can be permutation-invariant. Actually, our encoding shown in Figure 11 is not strict to obtain permutation-invariant representation since inter-layer dependencies are not included during the encoding process. We also tried to incorporate the order information during OPR encoding and we found similar results with the way we present in Figure 11, which we adopt in our experiments.

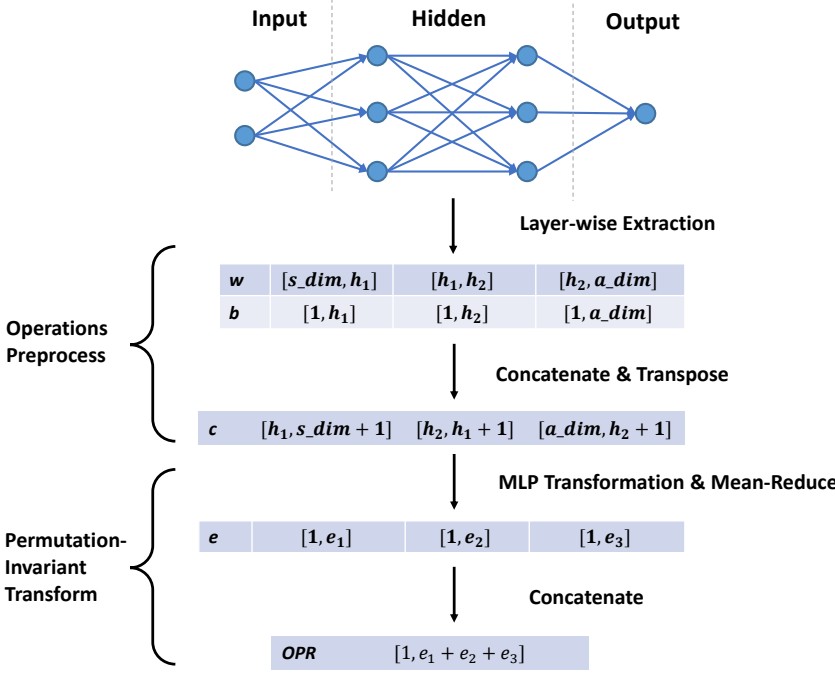

Figure 11: An illustration for policy encoder of Origin Policy Representation (OPR) for a two-layer MLP. $h_1, h_2$ denotes the numbers of hidden units for the first and second hidden layers respectively. The main idea is to perform permutation-invariant transformation for inner-layer weights and biases for each layer first and then concatenate encoding of layers.

**Towards more sophisticated RL policy that operates images.** Our proposed two policy representations (i.e., OPR and SPR) can basically be applied to encode policies that operate images, with the support of advanced image-based state representation. For OPR, a policy network with image input usually has a pixel feature extractor like Convolutional Neural Networks (CNNs) followed by a decision model (e.g., an MLP). With effective features extracted, the decision model can be of moderate (or relatively small) scale. Recent works on unsupervised representation learning like MoCo

(He et al., 2020), SimCLR (Chen et al., 2020), CURL (Srinivas et al., 2020) also show that a linear classifier or a simple MLP which takes compact representation of images learned in an unsupervised fashion is capable of solving image classification and image-based continuous control tasks. In another direction, it is promising to develop more efficient even gradient-free OPR, for example using the statistics of network parameters in some way instead of all parameters as similarly considered in (Unterthiner et al., 2020).

For SPR, to encode state-action pairs (or sequences) with image states can be converted to the encoding in the latent space. The construction of latent space usually involves self-supervised representation learning, e.g., image reconstruction, dynamics prediction. A similar scenario can be found in recent model-based RL like Dreamer (Hafner et al., 2020), where the imagination is efficiently carried out in the latent state space rather than among original image observations.

Overall, we believe that there remain more effective approaches to represent RL policy to be developed in the future in a general direction of OPR and SPR, which are expected to induce better value generalization in a different RL problems.

### D.3 DATA AUGMENTATION FOR SPR AND OPR IN CONTRASTIVE LEARNING

Data augmentation is studied to be an important component in contrastive learning in deep RL recently (Kostrikov et al., 2020; Laskin et al., 2020). Contrastive learning usually resorts to data augmentation to build postive samples. Data augmentation is typically performed on pixel inputs (e.g., images) problems (He et al., 2020; Chen et al., 2020). In our work, we train policy representation with contrastive learning where data augmentation is performed on policy data. For SPR, i.e., state-action pairs as policy data, there is no need to perform data augmentation since different batches of randomly sampled state-action pairs naturally forms positive samples, since they all reflect the behavior of the same policy. A similar idea can also be found in (Fu et al., 2020) when dealing with task context in Meta-RL.

For OPR, i.e., policy network parameters as policy data, it is unclear how to perform data augmentation on them. In this work, we consider two kinds of data augmentation for policy network parameters as shown in Figure 12. We found similar results for both random mask and noise corruption, and we use random mask as default data augmentation in our experiments.

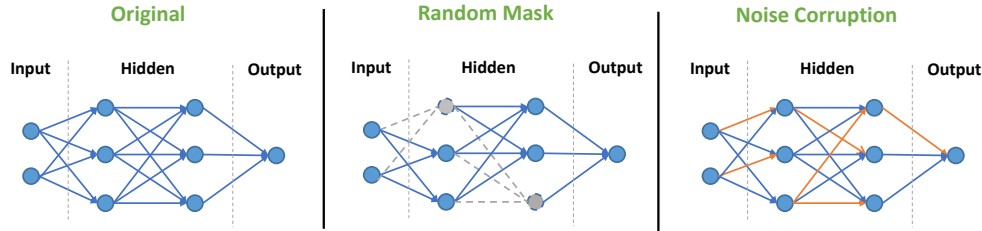

Figure 12: Examples of data augmentation on policy network parameters for Origin Policy Representation (OPR). *Left*: an example of original policy network. *Middle*: dropout-like random masks are performed on original policy network, where gray dashed lines represent the weights masked out. *Right*: randomly selected weights are corrupted by random noises, denoted by orange lines.

As an unsupervised representation learning method, contrastive Learning encourages policies to be close to similar ones (i.e., positive samples $\pi^+$) and to be apart from different ones (i.e., negative samples $\pi^-$) in policy representation space. The policy representation network is then trained with InfoNCE loss (Oord et al., 2018), i.e., to minimize the cross-entropy loss below:

$$\mathcal{L}_{\text{CL}} = -\mathbb{E}\left[\log \frac{\exp(\chi_\pi^T W \chi_{\pi^+})}{\exp(\chi_\pi^T W \chi_{\pi^+}) + \sum_{\pi^-} \exp(\chi_\pi^T W \chi_{\pi^-})}\right]$$

### D.4 PSEUDO-CODE OF POLICY REPRESENTATION LEARNING FRAMEWORK

The pseudo-code of the overall framework of policy representation learning is in Algorithm 3. Note that in Line 21, the positive samples $\chi_{\pi_i^+}$ is obtained from a momentum policy encoder (He et al.,

2020) with another augmentation for corresponding policy data, while negative samples $\chi_{\pi_i^-}$ are other policy embeddings in the same batch, i.e., $\chi_{\pi_i^-} \in B \backslash \{\chi_{\pi_i}\}$.

---

**Algorithm 3** A Framework of Policy Representation Learning

---

**Input:** policy dataset $\mathbb{D} = \{(\pi_i, \omega_i, \mathcal{D}_{\pi_i})\}_{i=1}^n$, consisting of policy $\pi_i$, policy parameters $\omega_i$ and state-action pairs $\mathcal{D}_{\pi_i} = \{(s_j, a_j)\}_{j=1}^m$

1: Initialize the policy encoder $g_\alpha$ with parameters $\alpha$
2: Initialize the policy decoder (or master policy) (network) $\bar{\pi}_\beta(a|s, \chi_\pi)$ for SPR and the weight matrix $W$ for ORP respectively
3: **for** iteration $i = 0, 1, 2, \ldots$ **do**
4:     Sample a mini-batch of policy data $\mathcal{B}$ from $\mathbb{D}$
5:     # Encode and obtain the policy embedding $\chi_{\pi_i}$ with SPR or OPR
6:     **if** Use OPR **then**
7:         **if** Use Contrastive Learning **then**
8:             Perform data augmentation on each $w_i \in \mathcal{B}$
9:         **end if**
10:        $\chi_{\pi_i} = g_\alpha^{\text{OPR}}(\omega_{\pi_i})$ for each $(\pi_i, \omega_i, \cdot) \in \mathcal{B}$
11:     **else if** Use SPR **then**
12:        $\chi_{\pi_i} = g_\alpha^{\text{SPR}}(B_i)$ where $B_i$ is a mini-batch of state-action pairs sampled from $\mathcal{D}_{\pi_i}$, for each $(\pi_i, \cdot, \mathcal{D}_{\pi_i}) \in \mathcal{B}$
13:     **end if**
14:     # Train policy encoder $g_\alpha$ in different ways (i.e., AUX or CL)
15:     **if** Use Auxiliary Loss **then**
16:        Sample a mini-batch of state-action pairs $B = (s_i, a_i)_{i=1}^b$ from $\mathcal{D}_{\pi_i}$ for each $\pi_i$
17:        Compute the auxiliary loss, $\mathcal{L}_{\text{Aux}} = -\sum_{(s_i, a_i) \in B} \log \bar{\pi}_\alpha(a_i|s_i, \chi_{\pi_i})$
18:        Update parameters $\alpha, \beta$ to minimize $\mathcal{L}^{\text{Aux}}$
19:     **end if**
20:     **if** Use Contrastive Learning **then**
21:        Calculate contrastive loss, $\mathcal{L}_{\text{CL}} = -\sum_{\chi_{\pi_i} \in B} \log \frac{\exp(\chi_{\pi_i}^T W \chi_{\pi_i^+})}{\exp(\chi_{\pi_i}^T W \chi_{\pi_i^+}) + \sum_{\pi_i^-} \exp(\chi_{\pi_i}^T W \chi_{\pi_i^-})}$,
    where $\chi_{\pi_i^+}, \chi_{\pi_i^-}$ are positive and negative samples
22:        Update parameters $\alpha, W$ to minimize $\mathcal{L}^{\text{CL}}$
23:     **end if**
24:     # Train policy encoder $g_\alpha$ with the PeVFA approximation loss (E2E)
25:     Calculate the value approximation loss of PeVFA, $\mathcal{L}_{\text{Val}}$
26:     Update parameters $\alpha$ to minimize $\mathcal{L}_{\text{Val}}$
27: **end for**

---

### D.5   A Review of Related Works on Policy Representation/Embedding Learning

Recent years, a few works involve representation or embedding learning for RL policy (Hausman et al., 2018; Grover et al., 2018; Arnekvist et al., 2019; Raileanu et al., 2020; Wang et al., 2020; Harb et al., 2020). We provide a brief review and summary for above works below.

The most common way to learn a policy representation is to extract from interaction trajectories through action recovery (i.e., behavioral cloning). For Multiagent Opponent Modeling (Grover et al., 2018), a policy representation is learned from interaction episodes (i.e., state-action trajectories) through a *generative loss* and *discriminate loss*. Generative loss is the same as the action prediction auxiliary loss; discriminate loss is a triplet loss that minimize the representation distance of the same policy and maximize those of different ones, which has the similar idea of Contrastive Learning (Oord et al., 2018; Srinivas et al., 2020). Such opponent policy representations are used for prediction of interaction outcomes for ad-hoc teams and are taken in policy network for some learning agent to facilitate the learning when cooperating or competing with unknown opponents. More recently, in Hierarchical RL (Wang et al., 2020), a representation is learned to model the low-level policy through *generative loss* mentioned above. The low-level policy representation is taken in high-level policy to counter the non-stationarity issue of co-learning of hierarchical policies. Later, Raileanu et al. (2020) resort to almost the same method and the learned policy representation is taken

in their proposed PDVF. Along with a task context, the policy for a specific task can be optimized in policy representation space, inducing a fast adaptation in new tasks. In summary, such a representation learning paradigm can be considered as Surface Policy Representation (SPR) for policy data encoding (trajectories as a special form of state-action pairs) plus action prediction auxiliary loss (AUX) as we introduced in Sec. 4.

A recent work (Harb et al., 2020) proposes Policy Evaluation Network (PVN) to approximate objective function $J(\pi)$. We consider PVN as an predecessor of PDVF we mentioned above since offline policy optimization is also conducted in learned representation space in a single task after similarly well training the PVN on many policies. The authors propose *Network Fingerprint* to represent policy network. To circumvent the difficulty of representing the parameters directly, policy network outputs (policy distribution) under a set of *probing states* are concatenated and then taken as policy representation. Such probing states are randomly sampled for initialization and also optimized with gradients through PVN and policies, like a search in joint state space. In principle, we also consider this as a special instance of SPR, because network fingerprint follows the idea of reflecting the information of how policy can behave under some states. Intuitively from a geometric view, this can be viewed as using the concatenation of several representative cross-sections in policy surface (e.g., Figure 10) to represent a policy. For another view, one can imagine an equivalent case between SPR and network fingerprint, when state-action pairs of a deterministic policy are processed in SPR and a representation consisting of a number of actions under some key states or representative states is used in network fingerprint. Two potential issues may exist for network fingerprint. First, the dimensionality of representation is proportional to the number of probing states (i.e., $n|\mathcal{A}|$), where a dilemma exists: more probing states are more representative while dimensionality can increase correspondingly. Second, it can be non-trivial and even unpractical to optimize probing states in the case with relatively state space of high dimension, which induces additional computational complexity and optimization difficulty.

In another concurrent work (Faccio & Schmidhuber, 2020), a class of Parameter-based Value Functions (PVFs) are proposed. Instead of learning or designing a representation of policy, PVFs simply parse all the policy weights as inputs to the value function (i.e., Raw Policy Representation as also mentioned in our paper), even in the nonlinear case. Apparently, this can result in a unnecessarily large representation space which increase the difficulty of approximation and generalization. The issues of naively flattening the policy into a vector input are also pointed out in PVN (Harb et al., 2020).

Others, several works in Policy Adaptation and Transfer (Hausman et al., 2018; Arnekvist et al., 2019), Gaussian policy embedding representations are construct through *Variational Inference*.

### D.6    CRITERIA OF A GOOD POLICY REPRESENTATION

To answer the question: what is a good representation for RL policy ought to be? We assume the following criteria:

- *Dynamics*. Intuitively, a good policy representation should contain the information of how the policy influences the environment (dynamics and rewards).

- *Consistency*. A good policy representation should keep the consistency among both policy space and presentation space. Concretely, the policy representation should be distinguishable, i.e., different policies also differ among their representation. In contrast, the representation of similar polices should lay on the close place in the representation space.

- *Geometry*. Additionally, from the lens of policy geometry as shown in Appendix D.1, a good policy representation should be an reflection of policy geometry. It should show a connection to the policy geometry or be interpretable from the geometric view.

From the perspective of above criteria, SPR follows *Dynamics* and *Geometry* while OPR may render them in an implicit way since network parameters determine the nonlinear function of policy. Auxiliary loss for action prediction (AUX) is a learning objective to acquire *Dynamics*; Contrastive Learning (CL) is used to impose *Consistency*.

Based on the above thoughts, we hypothesize the reasons of several findings as shown in the comparison in Table 1. First, AUX naturally overlaps with SPR and OPR to some degree for *Dynamics* while CL is relatively complementary to SPR and OPR for *Consistency*. This may be the reason

why CL improves the E2E representation more than AUX in an overall view. Second, the noise of state-action samples for SPR may be the reason to OPR's slightly better overall performance than that of SPR (similar results are also found in our visualizations as in Figure 19).

Moreover, the above criteria are mainly considered from an unsupervised or self-supervised perspective. However, a sufficiently good representation of all the above properties may not be necessary for a specific downstream generalization or learning problem which utilizes the policy representation. A problem-specific learning signal, e.g., the value approximation loss in our paper (E2E representation), can be efficient since it is to extract the most relevant information in policy representation for the problem. A recent work Tsai et al. also studies the relation between self-supervised representation and downstream tasks from the lens of mutual information. Therefore, we suggest that a trade-off between good unsupervised properties and efficient problem-specific information of policy representation should be considered when using policy representation in a specific problem.

# E   COMPLETE BACKGROUND AND MORE

## E.1   REINFORCEMENT LEARNING

**Markov Decision Process.**   We consider a Markov Decision Process (MDP) defined as $\langle \mathcal{S}, \mathcal{A}, r, \mathcal{P}, \gamma, \rho_0 \rangle$ with $\mathcal{S}$ the state space, $\mathcal{A}$ the action space, $r$ the reward function, $\mathcal{P}$ the transition function, $\gamma \in [0, 1)$ the discount factor and $\rho_0$ the initial state distribution. A policy $\pi \in \Pi = P(\mathcal{A})^{|\mathcal{S}|}$, defines the distribution over all actions for each state. The agent interacts with the environment with its policy, generating the trajectory $s_0, a_0, r_1, s_1, a_1, r_2, ..., s_t, a_t, r_{t+1}, ...$, where $r_{t+1} = r(s_t, a_t)$. An RL agent seeks for an optimal policy that maximizes the expected long-term discounted return, $J(\pi) = \mathbb{E}_{s_0 \sim \rho_0, a \sim \pi} \left[ \sum_{t=0}^{\infty} \gamma^t r_{t+1} \right]$.

**Value Function.**   Almost all RL algorithms involve value functions (Sutton & Barto, 1998), which estimate how good a state or a state-action pair is conditioned on a given policy. The *state-value function* $v^\pi(s)$ is defined in terms of the expected return obtained through following the policy $\pi$ from a state $s$:

$$v^\pi(s) = \mathbb{E}_\pi \left[ \sum_{t=0}^{\infty} \gamma^t r_{t+1} | s_0 = s \right] \text{ for all } s \in \mathcal{S}.$$

Similarly, *action-value function* is defined for all state-action pairs as $q^\pi(s, a) = \mathbb{E}_\pi \left[ \sum_{t=0}^{\infty} \gamma^t r_{t+1} | s_0 = s, a_0 = a \right]$. Typically, value functions are learned through Monte Carlo (MC) or Temporal Difference (TD) algorithms (Sutton & Barto, 1998).

Bellman equations defines the recursive relationships among value functions. The *Bellman Expectation equation* of $v^\pi(s)$ has a matrix form as below (Sutton & Barto, 1998):

$$V^\pi = r^\pi + \gamma \mathcal{P}^\pi V^\pi = (\mathcal{I} - \gamma \mathcal{P}^\pi)^{-1} r^\pi, \tag{14}$$

where $V^\pi$ is a $|\mathcal{S}|$-dimensional vector, $\mathcal{P}^\pi$ is the state-to-state transition matrix $\mathcal{P}^\pi(s'|s) = \sum_{a \in \mathcal{A}} \pi(a|s) \mathcal{P}(s'|s, a)$ and $r^\pi$ is the vector of expected rewards $r^\pi(s) = \sum_{a \in \mathcal{A}} \pi(a|s) r(s, a)$. Equation 14 indicates that value function is determined by policy $\pi$ and environment models (i.e., $\mathcal{P}$ and $r$). For a conventional value function, all of them are modeled implicitly within a table or a function approximator, i.e., a mapping from only states (and actions) to values.

## E.2   AN UNIFIED VIEW OF EXTENSIONS OF CONVENTIONAL VALUE FUNCTION FROM THE VECTOR FORM OF BELLMAN EQUATION

Recall the vector form of Bellman equation (Equation 14), it indicates that value function is a function of policy $\pi$ and environmental models (i.e., $\mathcal{P}$ and $r$). In conventional value functions and approximators, only state (and action) is usually taken as input while other components in Equation 14 are modeled implicitly. Beyond state (and action), consider explicit representation of some of components in Equation 14 during value estimation can develop the ability of conventional value functions in different ways, to solve challenging problems, e.g., goal-conditioned RL (Schaul et al., 2015; Andrychowicz et al., 2017), Hierarchical RL (Nachum et al., 2018; Wang et al., 2020), opponent modeling and ad-hoc team (He & Boyd-Graber, 2016; Grover et al., 2018; Tacchetti et al., 2019), and context-based Meta-RL (Rakelly et al., 2019; Lee et al., 2020).

As discussed in Sec. 2.2, most extensions of conventional VFA mentioned above are proposed for the purpose of value generalization (among different space). Therefore, we suggest such extensions are derived from the same start point (i.e., Equation 14) and differ at the objective to represent and take as additional input explicitly of conventional value functions. We provide a unified view of such extensions below:

- Goal-conditioned RL and context-based meta-RL usually focus on a series of tasks with similar goals and environment models (i.e., $\mathcal{P}$ and $r$). With goal representation as input, usually a subspace of state space (Schaul et al., 2015; Andrychowicz et al., 2017), a value function approximation (VFA) can generalize values among goal space. Similarly, with context representation (Rakelly et al., 2019; Fu et al., 2020; Raileanu et al., 2020), values generalize in meta tasks.

- Opponent modeling, ad-hoc team (He & Boyd-Graber, 2016; Grover et al., 2018; Tacchetti et al., 2019) seek to generalize among different opponents or teammates in a Multiagent System, with learned representation of opponents. This can be viewed as a special case of value generalization among environment models since from one agent view, other opponents are part of the environment which also determines the dynamics and rewards. In multiagent case, one can expand and decompose the corresponded joint policy in Equation 14 to see this.

- Hierarchical RL is also a special case of value generalization among environment models. In goal-reaching fashioned Hierarchical HRL (Nachum et al., 2018; Levy et al., 2019; Nachum et al., 2019), high-level controllers (policy) issue goals for low-level controls at an abstract temporal scale, while low-level controls take goals also as input and aim to reach the goals. For low-level policies, a VFA with a given or learned goal representation space can generalize values among different goals, similar to the goal-conditioned RL case as discussed above. Another perspective is to view the separate learning process of hierarchical policies for different levels as a multiagent learning system. Recently, a work (Wang et al., 2020) follows this view and extends high-level policy with representation of low-level learning.

The common thing of above is that, they learn a representation of the environment (we call *external variables*). In contrast, we study value generalization among agent's own policies in this paper, which cares about *internal variables*, i.e., the learning dynamics inside of the agent itself.

**Relation between PeVFA Value Approximation and Context-based Meta-RL.** For a given MDP, performing a policy in the MDP actually induces a Markov Reward Process (MRP) (Sutton & Barto, 1998). One can view the policy and actions are absorbed in the transition function of MRP. A value function defines the expected long-term returns starting from a state. Therefore, different policies induces different MRPs and PeVFA value approximation can be considered as a meta prediction task. In analogy to context-based Meta-RL where a task context is learned to capture the underlying transition function and reward function of a MDP (i.e., task), one can view policy representation as the context of corresponding MRP, since it is the underlying variable that determines the transition function of MRPs.

# F  EXPERIMENTAL DETAILS AND COMPLETE RESULTS

## F.1  EXPERIMENTAL DETAILS

**Environment.** We conduct our experiments on commonly adopted OpenAI Gym[1] continuous control tasks (Brockman et al., 2016; Todorov et al., 2012). We use the OpenAI Gym with version 0.9.1, the mujoco-py with version 0.5.4 and the MuJoCo products with version `MJPRO131`. Our codes are implemented with Python 3.6 and `Tensorflow`.

**Implementation.** We use Proximal Policy Optimization (PPO) (Schulman et al., 2017) with Generalized Advantage Estimator (GAE) (Schulman et al., 2016) as our baseline algorithm. Recent works (Engstrom et al., 2020; Andrychowicz et al., 2020) point out code-level optimizations influence the performance of PPO a lot. For a fair comparison and clear evaluation, we perform no

---

[1]http://gym.openai.com/

code-level optimization in our experiments, e.g., state standardization, reward scaling, gradient clipping, parameter sharing and etc. Our proposed algorithm PPO-PeVFA is implemented based on PPO, which only differs at the replacement for conventional value function network with PeVFA network. Policy network is a 2-layer MLP with 64 units per layer and ReLU activation, outputting a Gaussian policy, i.e., a tanh-activated mean along with a state-independent vector-parameterized log standard deviation. For PPO, the convention value network $V_\phi^\pi(s)$ (VFA) is a 2-layer 128-unit ReLU-activated MLP with state as input and value as output. For PPO-PeVFA, the PeVFA network $\mathbb{V}_\theta(s, \chi_\pi)$ takes as input state and policy representation $\chi_\pi$ which has the dimensionality of 64, with the structure illustrated in Figure 7. We do not use parameter sharing between policy and value function approximators for more clear evaluation.

**Training and Evaluation.** We use Monte Carlo returns for value approximation. In contrast to convention VFA $V_\phi^\pi$ which approximates the value of current policy (e.g., Algorithm 1), PeVFA $\mathbb{V}_\theta(s, \chi_\pi)$ is additionally trained to approximate the values of all historical policies ($\{\pi_i\}_{i=0}^t$) along the policy improvement path (e.g., Algorithm 2). The policy network parameterized by $\omega$ is then updated with following loss function:

$$\mathcal{L}^{\text{PPO}}(\omega) = -\mathbb{E}_{\pi_{\omega^-}}\left[\min\left(\rho_t \hat{A}(s_t, a_t), \text{clip}(\rho_t, 1-\epsilon, 1+\epsilon)\hat{A}(s_t, a_t)\right)\right], \qquad (15)$$

where $\hat{A}(s_t, a_t)$ is advantage estimation of old policy $\pi_{\omega^-}$, which is calculated by GAE based on conventional VFA $V_\phi^\pi$ or PeVFA $\mathbb{V}_\theta(s, \chi_\pi)$ respectively, and $\rho_t = \frac{\pi_\omega(a_t, s_t)}{\pi_{\omega^-}(a_t, s_t)}$ is the importance sampling ratio. Note that both PPO and PPO-PeVFA update the policy according to Equation 15 and only differ at advantage estimation based on conventional VFA $V_\phi^\pi$ or PeVFA $\mathbb{V}_\theta(s, \chi_\pi)$. This ensures that the performance difference comes only from different approximation of policy values. Common learning parameters for PPO and PPO-PeVFA are shown in Table 2. For each iteration, we update value function approximators first and then the policy with updated values. Such a training scheme is used for both PPO and PPO-PeVFA. For evaluation, we evaluate the learning algorithm every 20k time steps, averaging the returns of 10 episodes. Fewer evaluation points are selected and smoothed over neighbors and then plotted in our learning curves below.

Table 2: Common hyperparameter choices of PPO and PPO-PeVFA.

| Hyperparameters for PPO & PPO-PeVFA | |
|---|---|
| Policy Learning Rate | $10^{-4}$ |
| Value Learning Rate | $10^{-3}$ |
| Clipping Range Parameter ($\epsilon$) | 0.2 |
| GAE Parameter ($\lambda$) | 0.95 |
| Discount Factor ($\gamma$) | 0.99 |
| On-policy Samples Per Iteration | 5 episodes or 2000 time steps |
| Batch Size | 128 |
| Actor Epoch | 10 |
| Critic Epoch | 10 |
| Optimizer | Adam |

**Details for PPO-PeVFA.** For PeVFA, the training process also involves value approximation of historical policies and learning of policy representation. Training details are shown in Table 3. PeVFA $\mathbb{V}_\theta(s, \chi_\pi)$ is trained every 10 steps with a batch of 64 samples from an experience buffer with size of 200k steps. Policy representation model is trained at intervals of 10 or 20 steps depending on OPR or SPR adopted. Due to 1k - 2k policies are collected in total in each trial, a relatively small batch size of policy is used. For OPR, Random Mask (Figure 12) is performed on all weights and biases of policy network except for the output layer (i.e., mean and log-std). For SPR, two buffers of state-action pairs are maintained for each policy: a small one is sampled for calculating SPR and the relatively larger one is sampled for auxiliary training (action prediction).

Table 3: Training details for PPO-PeVFA, including value approximation of historical policies and policy representation learning. CL is abbreviation for Contrastive Learning and AUX is for auxiliary loss of action prediction. In our experiments, grid search is performed for the best hyperparamter configuration regarding terms with multiple alternatives (i.e., {}).

| Value Approximation for Historical Policies | |
| --- | --- |
| Value Learning Rate | $10^{-3}$ |
| Training Frequency | Every 10 time steps |
| Batch Size | 64 |
| Experience Buffer Size | 200k (steps) |
| Policy Representation Learning | |
| Training Frequency | Every $\{10, 20\}$ time steps |
| Policy Num Per Batch | $\{16, 32\}$ |
| SPR $s, a$ Pair Num | $\{200, 500\}$ |
| CL Learning Rate | $\{10^{-3}, 5 \cdot 10^{-4}\}$ |
| CL Momentum | $\{5 \cdot 10^{-2}, 10^{-2}, 5 \cdot 10^{-3}\}$ |
| CL Mask Ratio for OPR | $\{0.1, 0.2\}$ |
| CL Sample Ratio for SPR | 0.8 |
| AUX Learning Rate | $10^{-3}$ |
| AUX Batch Size | $\{128, 256\}$ |

## F.2 COMPLETE LEARNING CURVES FOR TABLE 1

Corresponding to Table 1, an overall view of learning curves of all variants of PPO-PeVFA as well as baseline algorithms are shown in Figure 16. One can refer to Figure 13 for Question 1 and Figure 14, 15 for Question 2, for clearer comparisons.

## F.3 OTHER EXPERIMENTAL ANALYSIS

In Figure 17, we analyze the sensitivity of value generalization (experimental results of PPO-PeVFA) with respect to the number of historical policies that PeVFA is trained on. The results show that the performance of PPO-PeVFA is not sensitive to relatively large numbers (i.e., 200k, 500k, 1000k) of historical state-action samples (proportional to the number of recent policies) used. However, a small number of historical samples may induce less sufficient and stable training of PeVFA, thus results in a slightly less improvement over the vanilla PPO.

## F.4 VISUALIZATION OF LEARNED POLICY REPRESENTATION

To answer Question 3, we visualize the learned representation of policies encountered during the learning process.

**Visualization Design.** We record all policies on the policy improvement path during the learning process of a PPO-PeVFA agent. For each trial in our experiments in MuJoCo continuous control tasks, about 1k - 2k policies are collected. We run 5 trials and 5k - 12k policies are collected in total for each task. We also store the policy representation model at intervals for each trial, and we use last three checkpoints to compute the representation of each policy collected. For each policy collected during 5 trials, its representation for visualization is obtained by averaging the results of 3 checkpoints of each trial and then concatenating the results from 5 trials. Finally, we plot 2D embedding of policy representations prepared above through t-SNE (Maaten & Hinton, 2008) and Principal Component Analysis (PCA) in `sklearn`[2].

**Results and Analysis.** Visualizations of OPR and SPR learned in an end-to-end fashion in HalfCheetah-v1 and Ant-v1 are in Figure 18 and 19. We use different types of markers to distinguish policies from different trials to see how policy evolves in representation space from different random initialization. Moreover, we provide two views: performance view and process view, to

---

[2]`https://scikit-learn.org/stable/index.html`

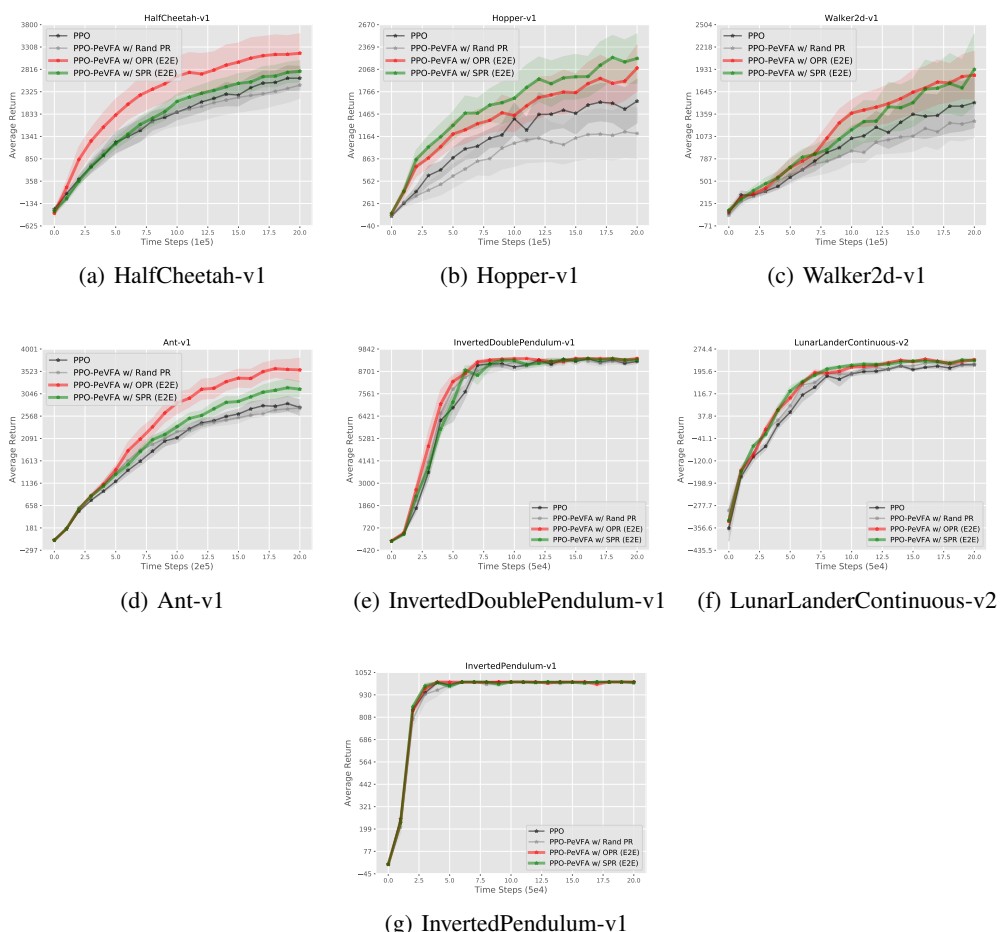

(a) HalfCheetah-v1

(b) Hopper-v1

(c) Walker2d-v1

(d) Ant-v1

(e) InvertedDoublePendulum-v1

(f) LunarLanderContinuous-v2

(g) InvertedPendulum-v1

Figure 13: Evaluations of PPO-PeVFA with end-to-end (E2E) trained OPR and SPR in MuJoCo continuous control tasks. The results demonstrate the effectiveness of PeVFA and two kinds of policy representation, answering the Question 1. The results are average returns and the shaded region denotes half a standard deviation over 10 trials.

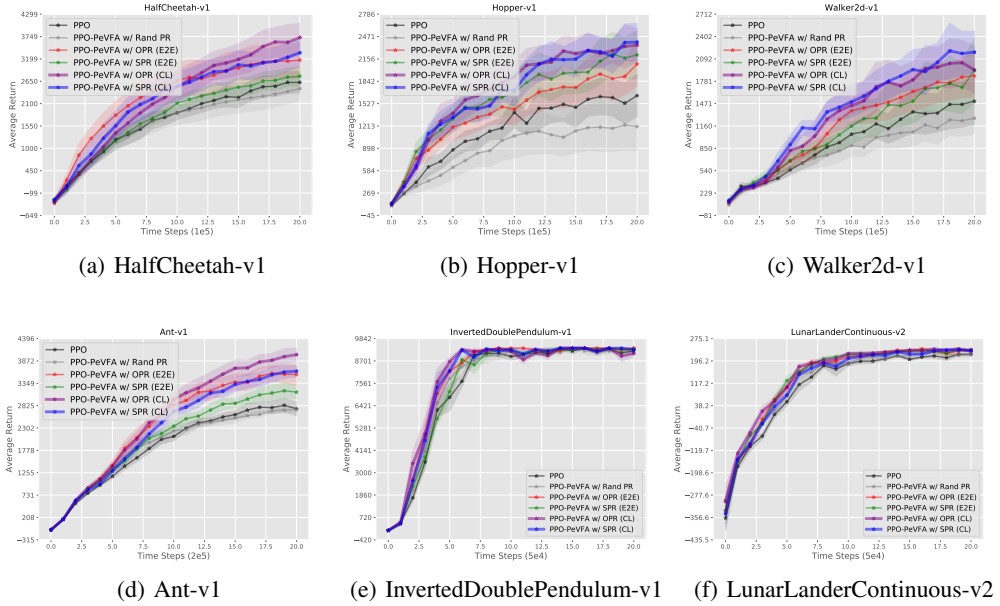

Figure 14: Evaluations of PPO-PeVFA with OPR and SPR trained through contrastive learning (CL) in MuJoCo continuous control tasks. The results are average returns and the shaded region denotes half a standard deviation over 10 trials.

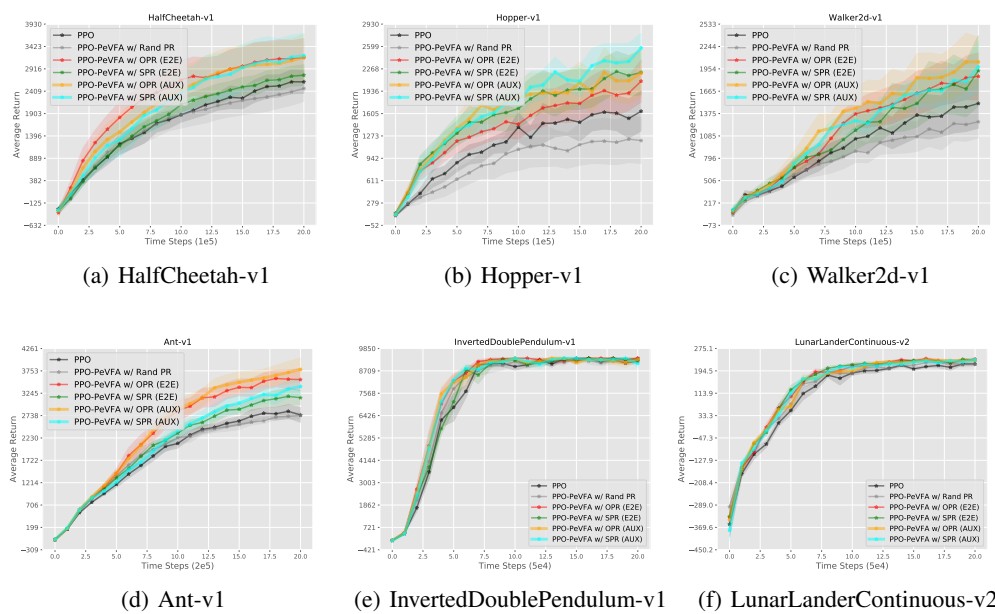

Figure 15: Evaluations of PPO-PeVFA with OPR and SPR trained through auxiliary loss of action prediction (AUX) in MuJoCo continuous control tasks. The results are average returns and the shaded region denotes half a standard deviation over 10 trials.

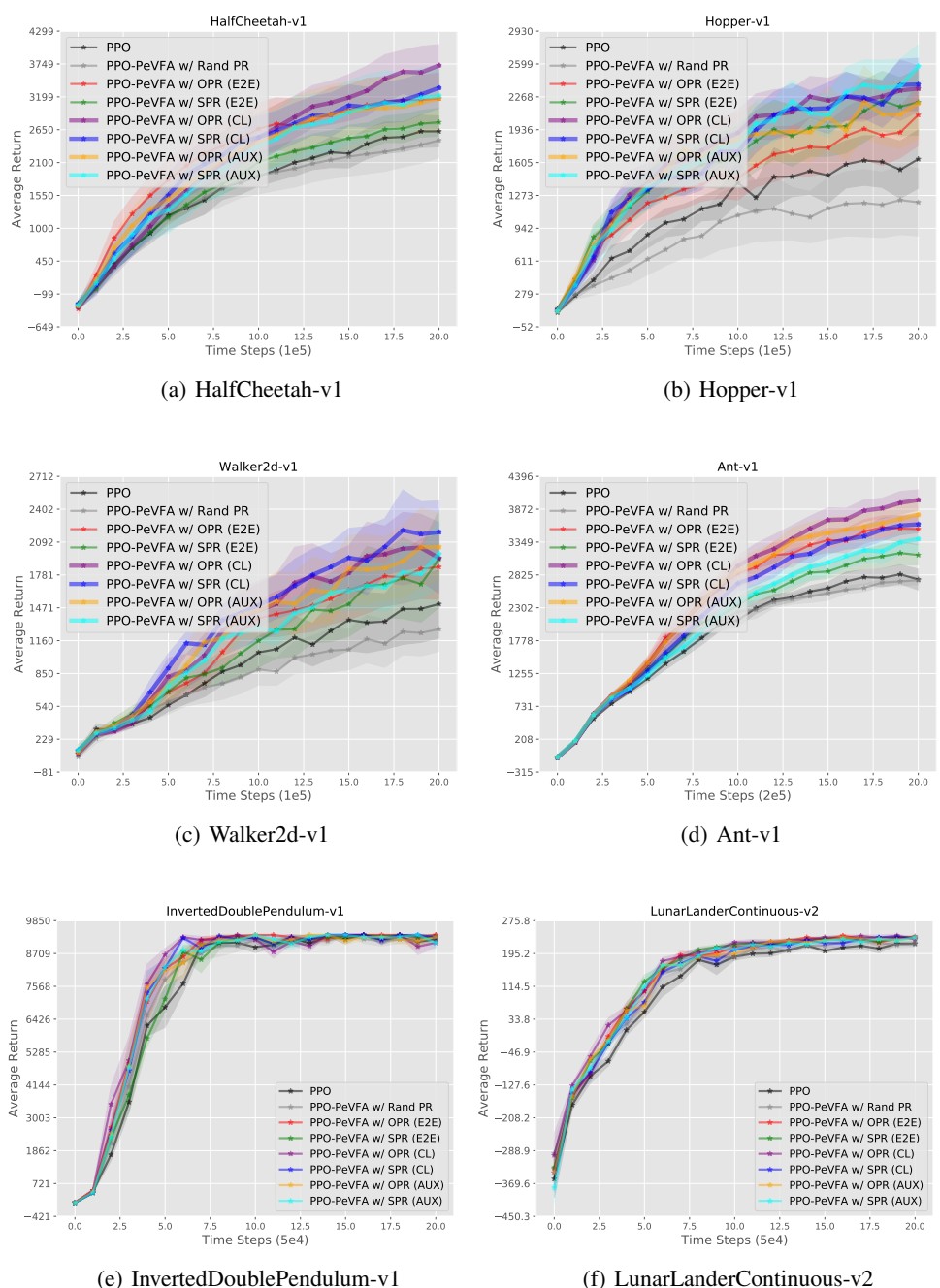

(a) HalfCheetah-v1

(b) Hopper-v1

(c) Walker2d-v1

(d) Ant-v1

(e) InvertedDoublePendulum-v1

(f) LunarLanderContinuous-v2

Figure 16: An overall view of performance evaluations of different algorithms in MuJoCo continuous control tasks. The results are average returns and the shaded region denotes half a standard deviation over 10 trials.

see how policies are aligned in representation space regarding performance and 'age' of policies respectively.

Visualization of OPR trained in end-to-end fashion is shown in Figure 18. From the performance view, it is obvious that policies of poor and good performances are aligned from left to right in t-SNE representation space and are aligned at two distinct directions in PCA representation space. An evolvement of policies from different trials can be observed in subplot (b) and (d). Thus, policies

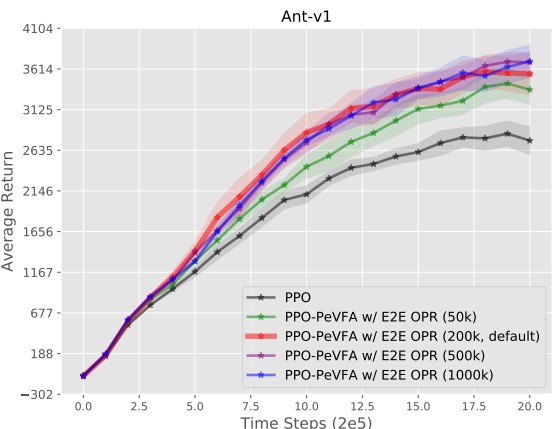

Figure 17: Results for PPO-PeVFA with End-to-End (E2E) OPR on Ant-v1 with respect to the number of historical policies that PeVFA is trained on. The setting varies at the numbers of historical samples generated by recent policies to different degrees, e.g., 50k denotes 50k steps of state-action samples from recent historical policies used for the training of PeVFA. The results are average returns and the shaded region denotes half a standard deviation over 10 trials.

from different trials are locally continuous; while policies are globally consistent in representation space with respect to policy performance. Moreover, we can observe multimodality for policies with comparable performance. This means that the obtained representation not only reflects optimality information but also maintains the behavioral characteristic of policy.

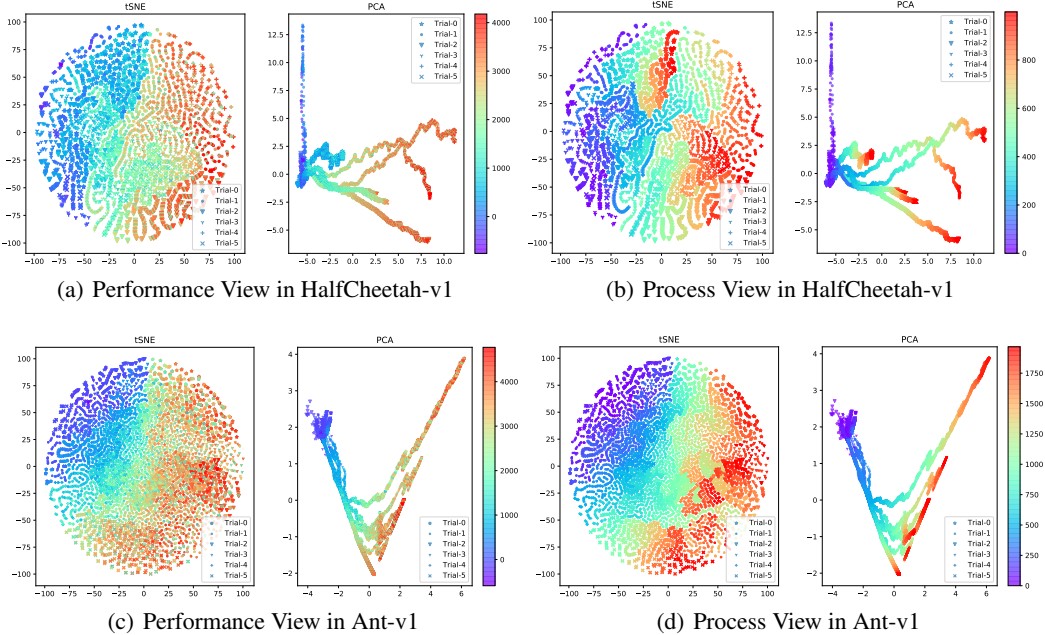

Figure 18: Visualizations of end-to-end (E2E) learned Origin Policy Representation (OPR) for policies collected during 5 trials (denoted by different kinds of markers). In total, about 6k policies are plotted for HalfCheetah-v1 (*a-b*) and 12k for Ant-v1 (*c-d*). In each subplot, t-SNE and PCA 2D embeddings are at left and right respectively. In performance view, each policy (i.e., marker) is colored by its performance evaluation (averaged return). In process view, each policy is colored by its corresponding iteration ID during GPI process.

Parallel to OPR, end-to-end trained SPR is visualized in Figure 19. A more obvious multimodality can be observed in both t-SNE and PCA space: policies from different trials start from the same region and then diverge during the following learning process. Different from OPR, SPR shows more distinction among different trials since SPR is a more direct reflection of policy behavior (*dynamics* property as mentioned in Sec. D.6). Another thing is, policies from different trials forms wide 'strands' especially in t-SNE representation space. We conjecture that it is because SPR is a more stochastic way to obtain representation as random selected state-action pairs are used.

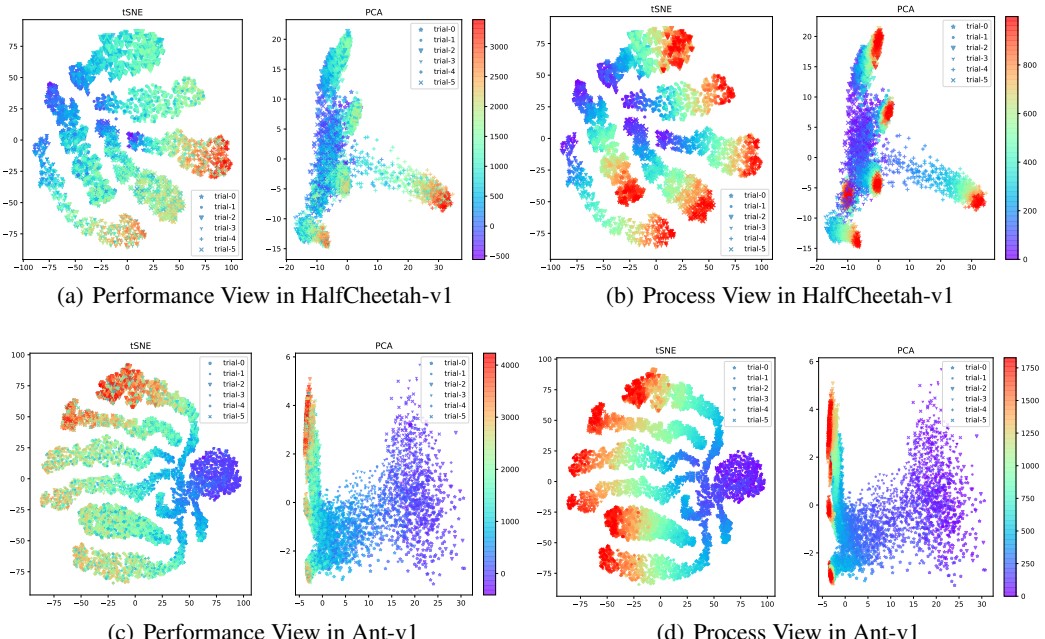

(a) Performance View in HalfCheetah-v1  (b) Process View in HalfCheetah-v1

(c) Performance View in Ant-v1  (d) Process View in Ant-v1

Figure 19: Visualizations of end-to-end (E2E) learned Surface Policy Representation (SPR) for policies collected during 5 trials (denoted by different kinds of markers). In performance view, each policy (i.e., marker) is colored by its performance evaluation (averaged return). In process view, each policy is colored by its corresponding iteration ID during GPI process.

