# OpenReview forum: "What About Taking Policy as Input of Value Function: Policy-extended Value Function Approximator"
_ICLR.cc/2021/Conference — Reject_

### Official Review · AnonReviewer4 · 2020-10-26
**Towards value function conditioned on the policy**

**Rating:** 7
**Confidence:** 3

**Review:**

The paper conditions the value function on a representation of the policy. The representation can be based on a batch of state-action pairs or based on the policy filters. When conditioning on the representation of a new policy, the value function can better approximate the value of the new policy. Experiments show benefits on continuous control tasks.

As mentioned in the paper, conditioning the value function on the policy was considered by some people before. It was unclear how to represent the policy. This paper makes it work and demonstrates clear benefits.
The policy representation can be still improved, and this paper can encourage people to try better representations.

The paper explains the ideas well.
I would like to see a more detailed description of the Surface Policy Representation (SPR). The Appendix D.5 is missing the review of the related works.

Pros:
- The value conditioned on the policy makes sense.
- Proposed end-to-end training, contrastive learning and auxiliary distillation to train the policy embedding.
- Tested raw policy representation, Surface Policy Representation and Origin Policy Representation.
- Demonstrated benefits in experiments.
- Visualization of the learned embeddings.

Cons:
- It is unclear how to practically encode policy representations for policies operating on images.

Minor typos:
- String "which beyond" should probably be "which are beyond".
- String "together with PeVFA end-to-end" should be "together with PeVFA is end-to-end".
- String "policies along the policy improvement path naturally provides" should be "policies along the policy improvement path naturally provide".
- String "Use PPR" should probably be "Use SPR".

---

> ### Author Response · Authors · 2020-11-21
> **Discussion on encoding policy representations for policies operating on images**
>
> We appreciate the reviewer’s valuable comments. In the revision version updated, we added a detailed description of how to encode state-action pairs for SPR in Appendix D.2 and the training details of SPR in Appendix F.1. The complete review of policy representation learning can be found in Appendix D.5 now, as a remedy to our mistake made in the previous submission (as mentioned in the $\textit{overall response}$). We thank the reviewer for pointing out the typos which have been already revised as suggested.
>
>
> Our proposed two policy representations (i.e., OPR and SPR) can basically be applied to encode policies that operate images, with the support of advanced image-based state representation.
> For OPR, a policy network with image input usually has a pixel feature extractor like Convolutional Neural Networks (CNNs) followed by a decision model (e.g., an MLP). We assume that with effective features extracted, the decision model can be of moderate (or relatively small) scale which is within the feasible extent of our proposed OPR. Recent works on unsupervised representation learning like MoCo [1], SimCLR [2], CURL [3] also show that a linear classifier or a simple MLP which takes compact representation of images learned in an unsupervised fashion is capable of solving image classification and image-based continuous control tasks. In another direction, we plan to develop OPR with more potential approaches, for example using the statistics of network parameters in some way instead of all parameters as similarly considered in [4].
> For SPR, to encode state-action pairs (or sequences) with image states can be converted to the encoding in the latent space. The construction of latent space usually involves self-supervised representation learning, e.g., image reconstruction, dynamics prediction. A similar scenario can be found in recent model-based RL like Dreamer [5], where the imagination is efficiently carried out in the latent state space rather than among original image observations.
>
> Finally, we consider that there remain more effective approaches to represent RL policy to be developed in the future in a general direction of OPR and SPR, which are expected to induce better value generalization in a different RL problems.
>
>
> ---------------------------------------------
>
> Reference:
>
> [1] Ting Chen, Simon Kornblith, Mohammad Norouzi, Geoffrey E. Hinton: A Simple Framework for Contrastive Learning of Visual Representations. CoRR abs/2002.05709 (2020)
>
> [2] Kaiming He, Haoqi Fan, Yuxin Wu, Saining Xie, Ross B. Girshick: Momentum Contrast for Unsupervised Visual Representation Learning. CVPR 2020: 9726-9735
>
> [3] Aravind Srinivas, Michael Laskin, Pieter Abbeel: CURL: Contrastive Unsupervised Representations for Reinforcement Learning. CoRR abs/2004.04136 (2020)
>
> [4] Thomas Unterthiner, Daniel Keysers, Sylvain Gelly, Olivier Bousquet, Ilya O. Tolstikhin: Predicting Neural Network Accuracy from Weights. CoRR abs/2002.11448 (2020)
>
> [5] Danijar Hafner, Timothy P. Lillicrap, Jimmy Ba, Mohammad Norouzi: Dream to Control: Learning Behaviors by Latent Imagination. ICLR 2020

---

> > ### Comment · AnonReviewer4 · 2020-11-22
> > **Response to authors**
> >
> > Thank you for the informative response.

---

### Official Review · AnonReviewer1 · 2020-10-27
**Interesting idea but the clarity and experiments can be improved**

**Rating:** 5
**Confidence:** 4

**Review:**

Summary

The paper proposes to learn a value function that takes as input both a state and a policy embedding (PeVFA), which is used to design a new version of a generalized policy iteration (GPI) algorithm, named PPO-PeVFA. The authors also introduce a new way of learning policy embeddings and show superior results relative to vanilla RL (PPO) on several MuJoCo tasks.


Strengths

I found the proposed idea to be novel and interesting. Considering the value of multiple policies is an interesting and neglected line of work, which could prove useful in many ways for RL, so I commend the authors for taking a step in this direction. I also particularly liked the theoretical analysis, as well as the experiments supporting the claim for local and global generalization. However, I believe there are a number of issues with the experiments and clarity of the paper, which I would like to see addressed.


Weaknesses

My main concern about this paper is the significance of the results and the empirical evaluation. The performance gain from using PeVFA does not seem very significant. After looking carefully at Table 1 and Figure 11 in the appendix, it seems like PPO-PeVFA is typically within a standard deviation of PPO. In addition, some of the reported results for PPO are much lower than the ones reported in the original PPO paper (Schulman et al. 2017). For example, on Hopper-v1 the reported numbers after 1M training sweets are 1600 and 2200, ,while for Walker2d-v1 they are 1500 and 3000 for the authors’ implementation and the original one, respectively.

In addition, I found the description of PPO-PeVFA to be quite confusing, so I think it can benefit from more details about the training procedure. Does the policy network share any parameters with the PeVFA network? Usually, parameters are shared between the policy and value function to improve learning, but it doesn’t seem like this is the case with PPO-PeVFA. Typically the policy and value function are updated at the same time in PPO but from Algorithm 2, it seems like before a policy update, you first update the PeVFA network with data from the current policy and then use the updated value to update the policy using the PG loss. Is my understanding correct? If this is the case, then it seems like the method is using some kind of privileged information about the current policy so an ablation that uses the same training scheme of first updating the value function (without conditioning on a policy) using the Monte Carlo rollouts from the current policy and then updating the policy should be included.

I also think it would be useful to include comparisons with other baselines that achieve strong results on MuJoCo such as SAC or TD3 or alternatively show that PeVFA can improve over them as well.

Do you have an intuition for why the difference between PeVFA and VFA is more significant for InvertedPendulum compared to Ant (and the other MuJoCo tasks) as illustrated in Figure 3b?
I’ve found these plots slightly worrying because it seems like the value approximation gains are rather small for most of these tasks. It would be interesting to better understand the relation between these value generalization errors and the performance improvement.

Have you looked into the sensitivity of PeVFA’s generalization with respect to the set (and number) of policies it is trained on? One can imagine training it on the entire history of policies during training or on a subset of these (perhaps the most recent ones). It would be useful to analyze how this choice affects the results.

Have you tried combining the contrastive loss with the auxiliary loss for action action prediction? Is this better than either of them or not?

Can you provide some quantitative metrics for the results in Figure 3a such as the MSE and ranking loss for the train and test values?

Why doesn’t Table 1 include results for InvertedPendulum? Can you add standard deviation for PPO and Ran PR?


Minor Points

Typos:
“which are beyond the scope of this paper” instead of “which beyond…” in section 3.3
“In the last section” instead of “in last…” in section 4
“To answer the above questions” instead of “...answer above...” in section 5


Recommendation

While the proposed idea is interesting and novel, I believe more work is needed for publication (particularly in the empirical evaluation and algorithm description), so I lean towards rejection at this stage.

---

> ### Author Response · Authors · 2020-11-21
> **Responses to concerns and questions on experimental details**
>
> We appreciate the reviewer’s valuable and constructive comments. We apologize that our mistake made in the first submission added unnecessary difficulty of evaluating our work due to the missing of algorithm and experiment details. A revision has been uploaded along with some clarifications as in the $\textit{Overall Response}$.
>
>
> $\textbf{Q1:}$ Experimental details and results
>
> 1) $\textbf{About PPO baseline.}$ In our experiments, we implement the algorithm core of original PPO [1] (vanilla PPO) and do not use code-level optimization (e.g., state normalization, reward scaling and etc.) which are demonstrated to impact the learning performance of PPO a lot in recent works on empirical studies [2,3]. This is for a clear and reliable evaluation of PPO-PeVFA, which prevents the interference of potential implementation-level variants. Experimental details can be seen in Appendix F.1 in the revision.
> In fact, the standard implementation of PPO [1] in OpenAI Baseline contains many non-trivial optimizations that are not (or only barely) described in its corresponding paper, as verified in [2,3]. Our vanilla PPO shows reasonable results on HalfCheetah-v1 and supervising results on Ant-v1 which are higher than those reported in other papers. We verified that our vanilla PPO with merely state normalization can achieve 2355.89 ($\pm$ 288.71) and 3110.06 ($\pm$ 129.11) averaged over 6 trials with current hyperparameters as shown in Table 2. Moreover, we referred to the large-scale empirical study work [2] and re-checked the validity of our vanilla PPO.
> 2) $\textbf{About parameter sharing.}$ We do not use parameter sharing between the policy and value function for both PPO and PPO-PeVFA. (It is also found that such a separation leads to better performance than parameter sharing in recent large-scale study on PPO [2].)
> 3) $\textbf{About training scheme.}$ PPO and PPO-PeVFA follow the same training scheme: at each iteration $t$, we train the value function and then update the policy, corresponding to the $\textit{policy evaluation}$ and $\textit{policy improvement}$ in GPI. (In practice, we also found that the order of value function and policy training showed no apparent influence on the results in our experiments.)
>
>
> $\textbf{Q2:}$ Intuitions on the difference between value approximation losses of PeVFA and VFA (i.e., generalization performance of PeVFA) in Figure 3b and Figure 8
>
> We think that the quantity of value approximation loss is related to several factors of the environment such as the reward scale, the extent of policy change, the complexity of underlying solution (e.g., value function space) and some others. A common observation across almost all results in Figure 8 is that the larger the extent of policy change (see the regions with a sheer slope on green curves), the higher the losses of conventional VFA tend to be (see the peaks of red curves), where the generalization tends to be better and more significant (see the blue curves). Since InvertedPendulum-v1 is a simple task while the complexity of the solution for Ant-v1 is higher, the difference between value approximation losses of PeVFA and VFA is more significant at the regions with fast policy improvement.
>
> Besides, the Raw Policy Representation (RPR) we used in Figure 3b and Figure 8 does not necessarily induce a smooth and efficient policy representation space, among which policy values are easy to generalize and optimize. Thus, RPR may be sufficient for a good generalization in InvertedPendulum-v1 but may be not in Ant-v1. This is also a reason why we study policy representation learning in this work.
>
>
> $\textbf{Q3:}$ PeVFA with TD value approximation (e.g., TD3, SAC)
>
> In this paper, we propose PPO-PeVFA as a representative instance of re-implementing DRL algorithms with PeVFA. Our purpose is to demonstrate that the local generalization introduced by PeVFA can improve conventional GPI and practical learning performance.
>
> Our theoretical results and algorithm 2 proposed under the general policy iteration (GPI) paradigm are suitable for TD value estimation as well. We plan to study PeVFA with TD learning (e.g., TD3 and SAC) in the future as mentioned in Sec.6.

---

> > ### Author Response · Authors · 2020-11-21
> > **(Continued)**
> >
> > $\textbf{Q4:}$ Sensitivity of value generalization (experimental results of PPO-PeVFA) with respect to the number of historical policies that PeVFA is trained on
> >
> > The results of PPO-PeVFA with different numbers of historical policies used for training is provided in Figure 17 in the revision updated. The setting varies at the numbers of historical state-action samples generated by recent policies to different degrees.
> > We can observe that PPO-PeVFA shows similar results for relatively large numbers (i.e., 200k, 500k, 1000k) of historical state-action samples (proportional to the number of recent policies) used. However, a small number of historical samples may induce less sufficient and stable training of PeVFA, thus results in a slightly less improvement over the vanilla PPO.
> >
> >
> > $\textbf{Q5:}$ Contrastive Learning (CL) and auxiliary loss of action prediction (AUX) for policy representation training
> >
> > In the revision, we also provided some further thoughts on CL and AUX under several criteria proposed by us in Appendix D.6. In our opinion, CL and AUX are to impose the $\textit{Consistency}$ property and the $\textit{Dynamics}$ property for policy representation respectively (with some underlying overlaps). We think a combination of CL and AUX can induce better performance though a proper weight is needed to balance the two kinds of losses with different scales.
> >
> >
> > $\textbf{Q6:}$ Other additional experimental results
> >
> > InvertedPendulum-v1 is a trivial task which can be easily solved by even a small policy network as in Figure 3b and Figure 9 (newly added). We did not see an apparent improvement for PPO-PeVFA in InvertedPendulum-v1 as the results updated in Figure 13g.
> > For empirical evidence of the global generalization (Figure 3a), the average MSE on training policy set is 2.909 and that on testing policy set is 4.155. We added the results in Sec. 3.2 as suggested. Table 1 is also updated with error bars for PPO and Ran PR as suggested.
> >
> > -------------------------
> >
> > Reference:
> >
> > [1] John Schulman, Filip Wolski, Prafulla Dhariwal, Alec Radford, Oleg Klimov: Proximal Policy Optimization Algorithms. CoRR abs/1707.06347 (2017)
> >
> > [2] Marcin Andrychowicz, et al.: What Matters In On-Policy Reinforcement Learning? A Large-Scale Empirical Study. CoRR abs/2006.05990 (v1) (2020)
> >
> > [3] Logan Engstrom, Andrew Ilyas, Shibani Santurkar, Dimitris Tsipras, Firdaus Janoos, Larry Rudolph, Aleksander Madry: Implementation Matters in Deep RL: A Case Study on PPO and TRPO. ICLR 2020

---

> > > ### Comment · AnonReviewer1 · 2020-11-24
> > > **Reviewer Response**
> > >
> > > Thank you for the clarifications and for updating the paper accordingly.
> > >
> > > Regarding the PPO implementation, while I understand your reasoning, it seems like the tricks that are standard practice such as state / reward normalization etc. are orthogonal to the proposed method so I would expect PeVFA to show similar gains with those tricks relative to PPO, unless the source of gains is different and somehow not entirely orthogonal to those other optimization tricks. I would still find that to be a better experimental setup given that it is standard practice to use such tricks and the absolute results would be easier to compare with other works using MuJoCo for evaluation. But I am willing to accept the argument that the use of such tricks is entirely orthogonal to PeVFA and the results should still stand in that setting.
> > >
> > > However, I am still worried about the significance of the results since PeVFA's performance is within a standard deviation of that of PPO in most cases. At the very least, I suggest not claiming (throughout the paper as well as in the abstract) that PeVFA "significantly outperforms" PPO since I do not find this to be accurate based on the empirical results presented in the paper. A more accurate statement would be that it is "comparable or slightly better in most cases". It is also a good practice to report 1 full standard deviation rather than half.
> > >
> > > Again, I think the idea is novel, interesting and could be valuable for the community, but I also believe the paper needs further improvements in order to be ready for publication.  Specifically, I think the writing can be improved to more clearly explain the proposed approach -- I still find it quite difficult to read and understand all the implementation details. In addition, the claims made must absolutely be well supported by the empirical and theoretical results shown in the paper. Finally, I think the paper would be even stronger if you can show more significant gains from PeVFA over vanilla RL in a certain setting / task, but I do not think that is strictly necessary for publication.

---

> > > > ### Author Response · Authors · 2020-11-25
> > > > **About Absolute Results**
> > > >
> > > > We appreciate the reviewer’s quick response.
> > > >
> > > > Our results in Table 1 for PPO-PeVFA w/ CL OPR (without code-level optimization) are 3725 $\pm$ 348 in HalfCheetah-v1 and 4019 $\pm$ 162 in Ant-v1, about 2x to 4x higher than the results of PPO reported in TD3 (also half a std there) and SAC paper.

---

### Official Review · AnonReviewer2 · 2020-10-28
**Interesting idea, but few overclaims; algorithmic implementation not clear, with key things not mentioned clearly. Difficult to understand the true significance of the work.**

**Rating:** 5
**Confidence:** 4

**Review:**

This work proposes an interesting idea of using the policy as an input to the value function. This is an interesting idea, moving away from the conventional approaches, and proposing to evaluate the value of a policy, by taking the policy as an input itself. Overall, it is an interesting direction, but there are few major concerns I have about the work, including some overstated claims which are not backed up properly.

Comments :
- The proposed PeVFA framework is interesting, since the value function can take as input many different policies, providing a way for generalization among policies. Based on this, the key algorithmic contribution is to propose generalized policy improvement (GPI) which can guarantee policy improvements among many different policies following the generalized value functions. Previous works have considered taking goal states or other heuristics as input, often for exploration purposes. In contrast, this work suggests that taking policy as input might lead to better generalization, especially in transfer learning situations, as originally proposed by Successor Features (Barreto et al) proposing the GPI algorithm.
- I do not think, however, that this idea is completely novel. Previous work on policy evaluation networks (Harb et al; arxiv paper from earlier this year) also proposed a similar approach, and this paper does not properly compare or take account of similar ideas from literature. It would be useful if the authors can comment on how their approach is different from Harb et al., given that the framework is almost similar.
- The key idea is to propose value generalization among policies, such that the value fn of a policy can generalize to a new policy, as outlined in figure 2 and section 3.1. I have a major concern about Assumption 1 being made - I am not fully convinced whether this is a valid assumption for the contraction property to hold in this. I do not think it is a major bottleneck, but to state this assumption makes me question whether the learnt PeVFAs can at all be useful.
- Theorem 1 is well stated. However, the proof of theorem 1 is not properly justified, and makes the contribution less significant. It would be useful if the authors can expand more on this, especially how the Lipchitz assumptions from Nesterov et al.,  are valid in this case. Otherwise, theorem 1 itself does not add much value for the overall contribution.
- My major concern with this work is how are PeVFas actually implemted in practice. The authors seem to build up from PPO, which would require the policies to be separately parameterized too. In this case, it would mean that the policy network is fed as input to the value function network itself. How does this approach scale up when the policy network is large? More importantly, how are gradients computed in this case? It is not clear whether this sort of hypernetwork idea is applicable for practical implementation of this approach, and this seems to be a major unavoidable issue with the work. Similar issues also arise in Harb et al., which they describe as the "fingerprinting" step. However, I think this is a major bottleneck for any of these works. Although the idea itself has potential, but taking policy as input, and considering parameterized policies might not be the way to go for this? I wonder if this kind of approach may be more preferrable in value based settings? Can the authors comment on this? Figure 4 seems to justify this sort of network architecture, but is not convincing.
- The paper discusses on the representation learning aspect of the policy. I understand that this is useful, leading to the generalization aspect this work is trying to propose. However, the claims are not made clear and difficult to understand. Is this work suggesting that the policy representations learnt through this framework can help learn invariant representations, which are then further useful for generalization? Some of the experimental demonstrations seem to address on this - however, it is not clear to what extent the invariant representations are indeed meaningful? As a reader, I was expecting the authors to compare and discuss experimentally with successor features and variants, for the GPI algorithm? However, there seem to be no discussions on those?
- Empirically, it is not clear what the paper is trying to propose. The authors claim to build up from PPO, but the experiments are not well justified for the generalization or transfer learning aspect? What would be the key takeaway from the experimental results? Table 1 is almost meaningless, given that the plots for these results in figure 11 in appendix shows marginal performance improvements. Experiments are only compared with PPO on control tasks, which does not have any generalization aspect per say? What are the authors trying to demonstrate in this case? I would suggest the authors propsoe their approach on simpler tasks, e.g four rooms domains and variants, and show how the proposed framework can be useful to transfer across tasks? Would that be a doable experiment that might add value to the contributions?
- Another minor issue is that it is not clear whether the algorithm can be implemented online or offline? How are the value functions trained, when the policy is used for rollouts? For example, if policy 1 is used for rollouts to compute the value functions, and doing a policy improvement step, how are these samples re-used when policy 2 is used as input to the value function network? I assume this will make the approach off-policy, since the rollouts under one policy would induce distribution shift compared to the other policy? I am not sure I fully understand how this algorithm can be implemented in practice, and what are the major drawbacks? Can the authors comment on the major difficulties of practical implementation of this idea? It would perhaps be useful if the pseudo-code of the algorithm can be included?

Overall, I think it is an interesting idea; but there are major concerns I have about the work which seem to be unaddressable? My main concern is with the practical implementation of this algorithm (I think I understand the intuitive and overall motivation for this, and why doing this might be useful). The paper claims to take this approach for the first time. However, there seems to be a prior work on a similar idea, which is not addressed in this work either. It would be useful if the authors can clarify on some of these doubts, for me to fully understand the practical significance of this work.

---

> ### Author Response · Authors · 2020-11-21
> **Response to concerns on experiments/implementation and clarification for the misunderstanding**
>
> We appreciate the reviewer’s detailed comments. After reading the comments, we apologize for our mistake (as stated in the $\textit{Overall Response}$) and we felt that the reviewer may have some misunderstanding on the main points of our paper to be evaluated. A few concerns can be addressed by the clarifications in $\textit{Overall Response}$ and we will also clarify all the detailed concerns and questions mentioned by the reviewer in the following.
>
>
> $\textbf{Q1:}$ The focus of this work
>
> The focus of this paper is the $\textit{local generalization}$, i.e., a scenario where we aim to utilize the value generalization among policies to improve the learning process in a single task. For this, we derive the condition of beneficial value generalization induced by PeVFA along the policy improvement path (Corollary 2) and provide empirical evidence for it in Figure 3b. Based on this, we then propose $\textit{GPI with PeVFA}$ (Figure 1 and Algorithm 2) and policy representations (Sec. 4) for a practical implementation. Our experimental results (Table 1 and Figure 13-16) are to show PPO-PeVFA that follows GPI with PeVFA is more efficient than the vanilla PPO that follows conventional GPI in several standard continuous control tasks.
>
> Therefore, our focus (i.e., local generalization) is not ‘transfer learning’ and ‘transfer across tasks’ as mentioned in the reviewer’s comments (although there is a potential for the $\textit{global generalization}$ to be further developed in transfer learning). We think this is important to understand and evaluate our work.
>
>
> $\textbf{Q2:}$ Comparison with PVN
>
> We thank the reviewer for pointing out the work of Policy Evaluation Network (PVN) which also studies the value function variant with policy as input. A discussion on PVN is provided in the $\textit{Overall Response}$. We also added detailed comments on PVN in the revision uploaded in Sec. 2.2 and Appendix D.5.
>
> We summarize the key differences between PVN and our work in the following:
> 1) PVN is an approximation of RL learning objective (i.e., $J(\pi_{\theta})$) while PeVFA is an extension of value functions.
> 2) Network Fingerprint is proposed in PVN to circumvent the infeasibility of using network parameters directly. In contrast, we propose a framework to learn a compact representation of RL policies in different ways. In practice, we represent larger policy network (an MLP with 2 hidden layers of 64 units for each) in our paper than that in PVN (an MLP with 1 hidden layer of 30 units). A detailed review on Network Fingerprint can be found in Appendix D.5, including a view on its similarity to the general idea of SPR and several potential drawbacks regarding optimizing the probing states.
> 3) PVN is used in an offline RL setting (i.e., optimize policy with a given set of sub-optimal or inferior policy data). On the contrary, we focus on the local generalization scenario, i.e., a typical online RL setting.
> 4) We conduct experiments in representative continuous control tasks in OpenAI Gym, rather than toy environments like CartPole and Swimmer used in PVN.
> 5) We theoretically study the value generalization since value generalization is not necessarily benefit to learning.
>
>
> $\textbf{Q3:}$ Questions about implementation details
>
> 1) $\textbf{About scalability}$. We use a normal-scale policy network (i.e., an MLP with 2 hidden layers of 64 units for each) in the experiments (Sec. 5). We do not feed the parameters of policy network in PeVFA and use a compact policy representation (i.e., SPR or OPR). The scalability of SPR is independent with the size of policy network since it encodes policy representation from state-action pairs; for OPR, the encoding approach we proposed (as in Figure 11) is applicable to common MLPs in principle. A discussion about policy representation encoding for policies which operate images can be found in the response to Reviewer 4.
> 2) $\textbf{About gradients}$. For PPO-PeVFA, its policy gradients are the same as those of vanilla PPO (also described in the $\textit{Overall Response}$); the gradients for policy representation is illustrated in Figure 4.
> 3)  $\textbf{About value-based settings}$. For value-based settings where the value function itself induces an implicit policy, to our knowledge, it is not clear how to utilize value generalizations among such implicit policies.  It deserves further study yet it is currently beyond the scope of this paper.
> Overall, our proposed policy representation and GPI with PeVFA are practical to implement and be applied in common RL environments.

---

> > ### Author Response · Authors · 2020-11-21
> > **(Continued)**
> >
> > $\textbf{Q4:}$ Policy representation and Successor Features (SF)
> >
> > It is not straightforward to consider a comparison between PeVFA (or policy representation) and SF in our setting. SF is a representation of value function that decouples the dynamics of the environment from the rewards (typically a weight vector) [1]. It is different from policy representation. From the unified view from Bellman Equation we provided in Appendix E.2, policy representation captures the policy while SF captures the long-term interplay of policy and environmental dynamics (transition function).
> >
> > Since SF decouples the dynamics and rewards, it can be combined with different reward functions (e.g., reward vectors) thus induces generalization ability to be transferred across tasks. As mentioned above we focus on the learning in single task.
> >
> >
> > $\textbf{Q5:}$ Implementation of the algorithm (GPI with PeVFA)
> >
> > 1) $\textbf{About the training of PeVFA}$. A general form of GPI with PeVFA is shown in Algorithm 2, where comments may help are colored in grey. The ‘multiple policies’ here are not policies which are independently optimized in parallel. In contrast, they are all historical versions and the successive versions to obtain by optimizing current policy along the policy improvement path.
> > 2) $\textbf{About off-policy learning}$. Off-policy learning typically denotes value estimation or policy optimization of a target policy by the experiences or samples generated by a different behavior policy. One interesting thing is that the value approximation training of PeVFA on historical policies is not necessary to be off-policy (i.e., it can also be on-policy). Since PeVFA preserves the values of multiple policies, the value estimation can be trained at the same time for different policies. In practice, we train PPO-PeVFA in an on-policy fashion. An additional discussion on this can be found in Appendix C.2.
> >
> >
> > $\textbf{Q6:}$ Concerns on theoretical assumptions
> >
> > The contraction condition (Assumption 1) is commonly adopted in theoretical analysis on policy evaluation or value estimation in RL. The Lipschitz conditions (Assumption 2) are often used in works about generalization study of neural networks [2,3] and RL [4,5] as well.
> >
> > Indeed, it is not necessary for the assumptions to be valid in practice, especially in the case where nonlinear function approximation is considered. We derive the Corollary 2 based on these two assumptions and provide empirical evidence in Figure 3b for it. The decrease in average value approximation error after training in Figure 3b can also be the empirical evidence for our Assumption 1 to some extent.
> > We thank the reviewer for constructive suggestions and we will polish our theoretical analysis as suggested.
> >
> >
> > We look forward to further discussions to address the reviewer’s concerns if possible.
> >
> > -----------------
> >
> > Reference:
> >
> > [1] André Barreto, Will Dabney, Rémi Munos, Jonathan J. Hunt, Tom Schaul, David Silver, Hado van Hasselt: Successor Features for Transfer in Reinforcement Learning. NIPS 2017: 4055-4065
> >
> > [2] Huan Wang, Nitish Shirish Keskar, Caiming Xiong, Richard Socher: Identifying Generalization Properties in Neural Networks. CoRR abs/1809.07402 (2018)
> >
> > [3] Mahyar Fazlyab, Alexander Robey, Hamed Hassani, Manfred Morari, George J. Pappas: Efficient and Accurate Estimation of Lipschitz Constants for Deep Neural Networks. NeurIPS 2019: 11423-11434
> >
> > [4] Pierluca D'Oro, Wojciech Jaskowski: How to Learn a Useful Critic? Model-based Action-Gradient-Estimator Policy Optimization. NeurIPS 2020
> >
> > [5] Zhizhou Ren, Kefan Dong, Yuan Zhou, Qiang Liu, Jian Peng: Exploration via Hindsight Goal Generation. NeurIPS 2019: 13464-13474

---

### Official Review · AnonReviewer3 · 2020-10-28
**Issues: theory / experiments / related work**

**Rating:** 3
**Confidence:** 5

**Review:**

Summary:

The authors propose PeVFA: a value function able to evaluate the expected return of multiple policies. They do so by extending the conventional value function, allowing it to receive as input the parameter (or a representation) of the policy. The authors study the local generalization property of PeVFA, propose possible ways of encoding the policy parameters and compare traditional PPO with an extended version of PPO using PeVFA. While the idea of generalization among many policies is an interesting topic in RL, there are many theoretical and experimental issues that prevent acceptance. Moreover, the authors do not at all compare their approach to recent work which also uses value functions with policy parameters as input.

Review:

Below the major problems I found:

- State and action value functions receiving as input the parameters of a policy, which generalize to unseen policies were proposed in a work on Parameter-based Value Functions [1] (PVFs, see version v1 from 16 Jun 2020). Recent work on Policy Evaluation Networks (PENs) proposes policy embedding for a value function receiving as input the parameters of a policy [2]. These works introduce value functions which generalize to many policies. The authors must relate their Surface Policy Representation to the fingerprint mechanism in PENs and introduce their V and Q in the PVF framework.

- Since the state space is not finite, $f_{\theta}(\pi)$, the approximation loss of $V_{\theta}$ should be defined considering an expectation over the state space, which is weighted by a distribution over the states (e.g. the on-policy stationary distribution). Since the loss proposed does not include a distribution over the states, it is not clear if the contraction assumption should hold in expectation or for every possible state. The former would imply that all the results should be stated in a probabilistic framework; the latter would be false in a continuous state setting.

- The authors assume that the class of value function considered can achieve zero approximation error. Where is this assumption used in the proofs and why is it useful?

- Theorem 1 seems quite trivial. It is trivial that if the loss is decreasing in one point and it has some smooth properties, then there exists a close enough point such that the loss decreases also there. Assumption 2 should be stated in a clearer way, differentiating the cases of Lipschitz continuous function, Lipschitz continuous gradient and Lipschitz continuous Hessian more clearly.

- Assuming that $\gamma_g$ in Corollary 1 is lower than 1 during the training process is quite unreasonable, and the authors do not check if this holds in their experiments. I expect that for Corollary 1 to hold, the learning rate should be extremely small, thus preventing learning.

- In Appendix A.2, there is an additional proof assuming that also $f(\pi_2)$ is Lipschitz. The authors start from the bound in eq (8), which is strictly less tight than the assumption $f(\pi_1) \leq f(\pi_2)$. By assuming only $f(\pi_1) \leq f(\pi_2)$ one would get the same final condition for $f(\pi_1)$, except for the term L_0 which would be zero. Hence the condition could be even less restrictive.

- In Corollary 2 it is assumed that the sum of the losses over 2 consecutive policies is lower than the distance between the optimal value function of the two policies. Is there an interpretation of this assumption or is it just a technical requirement to complete the proof? Please discuss why this assumption should hold during the learning process.

- When the policy representation is learned, the problem of mapping the policy parameters to the expected return becomes nonstationary, i.e. the same policy representation over time would be optimally mapped onto different values. How is this addressed in the experiments and how does nonstationarity affect the theoretical claims?

- From the experiments it is not clear if a representation for the policy is necessary. I would have expected to see a comparison between PeVFA with raw policy representation (RPR) as input and PeVFA using the learned representations. The authors should provide strong evidence that RPR is not enough and the policy representation is needed. Furthermore, the policy used is very small (2 layers and 2 neurons per layer). I would expect to see benefits in using a policy representation when the policy is bigger (e.g. 2 hidden layers, 64 or 128 neurons per layer).

- The authors provide no details about the hyperparameters used and about how they tuned their methods. Policy representation learning methods are only in part explained. Without further details, it is difficult to assess if the experiments provided were fair, and it is not possible to reproduce the results.

- In appendix B.1. the transition function is not reported. The state includes sinusoidal terms that do not appear in the reward function. Why is it necessary for the agent to observe these terms? Are not just the x-y coordinate sufficient?

- In Figure 3 and 7 it is not clear why there is only one learning curve for the policy policy and 6 value functions losses in standard PPO.

- Algorithms 1 and 2 make almost no distinction in training when the update is on-policy or off-policy, because the authors consider PeVFA as just a replacement for standard value function when the algorithm is already derived. However, if PeVFA is introduced before deriving the algorithm, the derivation leads to different policy gradient theorems (see [1]). Please discuss this issue.

- Figure 8 represents $\pi(a|s)$ for each possible a, s. Therefore I would expect that for each s, the integral over A of the curve is 1. However, the area under the curve is much lower than 1. What is the explanation for this?

- Last page in the Appendix is truncated.

I think the most interesting contribution of this paper is the proposed policy representation. I would like to see a revised version published in the future. However, a lot of additional work is needed to address the aforementioned problems in the theory and the missing experimental evidence & implementation details & comparisons to very similar related work.

Minor:
- The title is grammatically incorrect.
- "with a limited parameter space" -> what does limited mean? Finite? Or that $\Theta$ is only a subset of the space of value functions?
- same for "unlimited"
- Appendix C.1 "Advantage Acotor-Critic" -> Advantage Actor-Critic


[1] Francesco Faccio and Juergen Schmidhuber. Parameter-based Value Functions. arXiv preprint arXiv:2006.09226v1, 2020.

[2] Jean Harb, Tom Schaul, Doina Precup, and Pierre-Luc Bacon. Policy evaluation networks. arXiv preprint arXiv:2002.11833, 2020.

********************

Edit: responses after rebuttal:

> The class of PVFs are developed to be applied and evaluated in the online learning setting.

This is not true. See for instance off-line and zero-shot learning experiments with PVFs that can learn new policies which do not interact with the environment and still show generalization.

> We think this may also be a reason to the insignificant improvement compared with their baselines in their experiments (their Figure 2), even the inferior results in simple tasks like InvertedPendulum and CartPole. In contrast, our proposed policy representations enable PeVFA to be compatible with normal-scale policy networks and we then demonstrate the superiority of PPO-PeVFA against PPO in standard continuous control task of OpenAI Gym

I would strongly discourage you from claiming this. PPO-PeVFA is built on top of PPO, so it is very hard for your algorithm to perform worse than the baseline. On the other hand, PVFs and PENs are novel algorithms that rely COMPLETELY on the prediction of the value function. It is expected that they might outperform baselines in some environments and be comparable or worse in others. The PSSVF proposed in the work on PVFs is outperforming the baseline ARS (Mania et al. 2018) in all environments but Reacher, even when the policy is a neural net with 2 layers and 64 neurons per layer.

About Q2:

1. If the authors are using the $L_{\infty}$ norm, then the results practically apply only to finite state MDPs. Indeed, this is the case for the papers [2,3,4] cited by the authors in the comment above. However, the experiments proposed in the paper deal with continuous state spaces, so the theoretical results are disconnected.

It is still not clear what the assumption in Corollary 2 means and if that assumption is met in practice. If the authors cannot justify the assumption, the Corollary should be removed. The novelty of Theorem 1 is not obvious, since it seems to be a quite trivial result from Mathematical Analysis.

> The results show that PeVFA consistently shows lower losses (i.e., closer to approximation target) across all tasks than convention VFA before and after policy evaluation along policy improvement path, which demonstrates Corollary 2

Again, is the assumption of Corollary 2 met here? If you have a logical rule (or corollary) of the form: assumption -> consequence, and you show that the consequence holds, this does not imply that he assumption is true!

One main argument in favor of PVFs is that even with a much bigger policy, PVFs are able to outperform the results of PENs. This suggests that Raw Policy Representation (RPR) can be a strong baseline and any work trying to introduce a different policy representation should at least compare to RPR. Without such a baseline it is difficult to assess the benefits of the proposed policy representations.

Note that PeVFAs are PVFs! The PVF formalism already accounts for all kinds of policy representations.

Of course, all of these relations should also be clarified in title and abstract.


(Mania et al. 2018): Horia Mania, Aurelia Guy, and Benjamin Recht (2018).  Simple random search of static linear policies is competitive for reinforcement learning.  In Advances in Neural Information Processing Systems,pp. 1800-1809, 2018.

---

> ### Author Response · Authors · 2020-11-23
> **Response to concerns/questions on theory/experiments/related work and clarification for the misunderstandings**
>
> We appreciate the reviewer’s detailed comments. We thank the reviewer for pointing out PVN (i.e., PEN) and PVFs which have a similar form of value function extensions. We have already added them in Sec. 2.2, Sec. 4 and Appendix D.5 in our revision updated, to complete and improve our paper.
>
> We apologize the unnecessary difficulties of reviewing this paper due to the incomplete appendix we mistakenly submitted. A complete appendix with sufficient details on experiments and implementations is provided in the revision we uploaded. Any further question on experimental details is welcome to make this work better.
>
> In the following, we will try to address the reviewer’s concerns and to clarify a few misunderstandings as possible. Further discussions are expected if necessary.
>
>
>
> $\textbf{Q1:}$ Comparison with related work (i.e., PVN, PVFs)
> In the $\textit{Overall Response}$, we summarized a few major differences between our work and PVN/PVFs. We emphasize a few important points to address the reviewer’s concern and clarify the infeasibility of comparing our algorithm with them.
> 1) For PVN, we found that it is demonstrated to be effective in an offline policy optimization setting, which is different from the local generalization scenario (i.e., typically online learning process as the experiments in Sec. 5). Please refer to the $\textbf{Q2}$ in the response to $\textit{Reviewer 2}$ for all key differences between PeVFA and PVN.
> 2) The class of PVFs are developed to be applied and evaluated in the online learning setting. However, they use raw policy parameters as inputs (i.e., Raw Policy Representation we use in Sec. 3.2 for demonstration); the approximation of PVFs and the policy optimization is conducted among the raw policy parameter space. In our evaluation experiments (Table 1, Figure 13-16), we use a normal-scale policy network (an MLP with 2 hidden layers with 64 units for each) with 4k+ to 10k+ parameters depending on the specific environment with states of different dimensionalities. It is not feasible to input a vector of so many parameters along with state (or state-action) into a normal-scale value function. Such a curse of dimensionality faced by PVFs is admitted at the conclusion of their original paper and also mentioned in Sec. 4 of PVN. We think this may also be a reason to the insignificant improvement compared with their baselines in their experiments (their Figure 2), even the inferior results in simple tasks like InvertedPendulum and CartPole.
> In contrast, our proposed policy representations enable PeVFA to be compatible with normal-scale policy networks and we then demonstrate the superiority of PPO-PeVFA against PPO in standard continuous control task of OpenAI Gym.
> 3) Moreover, the source code of both PVN and PVFs are not available.
>
>
>
> $\textbf{Q2:}$ Concerns on the theory
> 1) $\textbf{About the definition of}$ $f_{\theta}(\pi)$. We use $L_{\infty}$ norm as a distance metric for given two value function approximation over the state space, based on which we define the approximation error $f_{\theta}(\pi)$ of parameter $\theta$ for policy $\pi$. This is a commonly adopted metric when studying value estimation (policy evaluation), which can be seen in the standard policy evaluation algorithm (Sec. 4.1 in the 2nd edition of [1]) as well as in classic RL works like [2-4]. We conduct our theoretical analysis by following this natural choice.
> 2) $\textbf{About zero approximation error}$. We consider the zero approximation error for the convenience of our theoretically analysis. Since we study the value approximation for multiple policies in contrast to the previous work of single-policy evaluation, zero approximation error allows to avoid the case where the optimal parameters for the whole policy space $\Pi$ are different from the optimal parameters for one single policy $\pi$, i.e., $\theta^* = \arg\min_{\theta \in \Theta} F(\theta, \Pi)$ may be not the same $\theta_{\pi}^* = \arg\min_{\theta \in \Theta} \||V_{\theta}(\cdot,\pi) - v^{\pi}(\cdot) \||_{\infty} $.
>
>
> -----------
> Reference:
>
> [1] Richard S. Sutton, Andrew G. Barto: Reinforcement Learning: An Introduction. IEEE Trans. Neural Networks 9(5): 1054-1054 (1998)
>
> [2] Sham M. Kakade, John Langford: Approximately Optimal Approximate Reinforcement Learning. ICML 2002: 267-274
>
> [3] Rémi Munos: Error Bounds for Approximate Policy Iteration. ICML 2003: 560-567
>
> [4] Michail G. Lagoudakis, Ronald Parr: Least-Squares Policy Iteration. J. Mach. Learn. Res. 4: 1107-1149 (2003)

---

> > ### Author Response · Authors · 2020-11-23
> > **(Continued)**
> >
> > $\textbf{Q3:}$  Interpretation of Corollary 1 and 2
> >
> > Since previous works (e.g., PVN and PVFs) do not study the conditions when value generalization among policies can be beneficial to RL from a theoretical lens, we derive Corollary 1 and 2 for such conditions in both a general global generalization scenario and a specific local generalization scenario, to gain some useful insights.
> >
> > We derive Corollary 1 for a condition for value approximation contraction ($\gamma_{g} \le 1$) in the general two-policy case. This provides a few intuitive understandings as discussed in Remark 2. The empirical results in Figure 3a provide some evidence for such a condition in the global generalization scenario: the value approximation error on the unseen policies in testing set gradually decreases during the learning of PeVFA on the policies in training set, and it finally show comparable approximation performance on testing set.
> >
> > In the local generalization scenario, we derive a condition in Corollary 2 for the case where value generalization can be beneficial to the value approximation (i.e., policy evaluation) in GPI, i.e., closer approximation distance caused by a generalized start point. We then provide empirical evidence for it in Figure 3b. Based on this, we then propose $\textit{GPI with PeVFA}$ (Figure 1 and Algorithm 2) and expect the improved value estimation to improve the quality of policy improvement and thus the whole GPI process.
> >
> > Additionally, we provided more empirical results in Figure 9 in the revision, showing similar evidence of the value generalization under the same setting as Figure 3b and Figure 8 but with larger learning rate in {$1e^{-4}, 1e^{-3}, 5e^{-3}$}.
> >
> >
> > $\textbf{Q4:}$  Policy representation
> >
> > We use a normal-scale policy network with 2 hidden layers and 64 units for each in all the experiments in Sec. 5. In this case, the infeasibility of directly parsing all the network parameters as input and the necessity of efficient policy representation have been clarified in the previous question.
> >
> > Any learning problem that involves representation learning (e.g., state representation, context representation) can commonly
> > have non-stationarity due to representation changes. We think that the gradients from value approximation loss (as illustrated in Figure 4) do some favor to stabilize the representation, since it lets the representation capture the information specific to the learning problem. This is also commonly used in works that equip RL with representation learning (e.g., CURL [5], PEARL [6]). Besides, we found an appropriate training interval of policy representation works well in our experiments. Concretely, we perform policy representation training with CL or AUX losses every 10 or 20 time steps (see Table 3 in Appendix F.1). An over-frequent training of policy representation may not be preferred due to the potential overfitting and non-stationarity induced.
> >
> >
> > $\textbf{Q5:}$ Different policy gradients used in PVN/PVFs and PeVFA
> >
> > In this paper, our main point is to leverage the benefits of the local generalization to achieve better value estimation (approximation) which is expected to improve the whole GPI process in turn. As discussed in the $\textit{Overall Response}$,  we stick to conventional PGs and do not use GTPI for policy update which are adopted in PVN and PVFs. We are conservative about GTPI in this paper and we consider a study in the future on the integration of conventional PGs and GTPI.
> > Overall, PeVFA only helps in value estimation learning and we do not consider using the conventional PGs to be an issue.
> >
> >
> >
> > $\textbf{Q6:}$ Empirical evidence in Figure 3b and Figure 7 (i.e., Figrue 8 in current revision)
> >
> > In Figure 3b and Figure 8, we compare the value approximation error (i.e., losses) of conventional VFA (i.e., red curves, $\||V_{\theta_{t}}(\pi_t) - V_{\theta^*}(\pi_{t+1})\||$) and PeVFA (i.e., blue curves, $ f_{\theta_t}(\pi_{t+1}) = \||V_{\theta_{t}}(\pi_{t+1}) - V_{\theta^*}(\pi_{t+1})\||$) along the learning process of a small PPO policy (green curves), thus 4 loss curves and 1 performance curve in total. Note that PeVFA here acts only as a reference for the loss comparison but does not influence the policy update. This is to provide the empirical evidence for the local generalization of PeVFA, especially Corollary 2.
> >
> > In the revision uploaded, we added a more detailed discussion and analysis in both Sec. 3.2 and Appendix B.2 (colored by orange) for the ease of understanding the results of Figure 3b and Figure 8.
> >
> >
> > ----------------
> > Reference:
> >
> > [5] Aravind Srinivas, Michael Laskin, Pieter Abbeel: CURL: Contrastive Unsupervised Representations for Reinforcement Learning. CoRR abs/2004.04136 (2020)
> >
> > [6] Kate Rakelly, Aurick Zhou, Chelsea Finn, Sergey Levine, Deirdre Quillen: Efficient Off-Policy Meta-Reinforcement Learning via Probabilistic Context Variables. ICML 2019: 5331-5340

---

> > > ### Author Response · Authors · 2020-11-23
> > > **(Continued)**
> > >
> > > $\textbf{Q7:}$ About Appendix B.1
> > >
> > > The transition function of 2D Walker environment is revised to be complete as in the revision updated.
> > >
> > > In 2D Walker, we synthesize a lot of small policy network with 2 hidden layers and 2 units for each to build the policy set for the demonstration of the global generalization. The reason of using sinusoidal terms along with x-y coordinate as the input of policy network is, we found this increases the diversity of synthetic policies.
> > >
> > >
> > > $\textbf{Q8:}$ About policy geometry in Figure 8 (i.e., Figure 10 in current revision)
> > > We thank the reviewer for pointing out the mistake: we illustrated the two axes for states and actions in a reverse order for Figure 10a and 10b incorrectly. This has been revised in the revision updated. Moreover, more detailed clarification of generation of the policy geometries is added in the caption of Figure 10.
> > >
> > >
> > > $\textbf{Minor:}$
> > > 1) Truncated figures for visualization are revised and complete details and analysis for visualizations are provided in Appendix F.2.
> > > 2) We use the ‘limited/unlimited’ for the practical/ideal cases where imperfect/perfect estimation can be obtained with function approximation within parameter space $\Theta$. We revised this as suggested.

---

### Author Response · Authors · 2020-11-16
**Overall response about experimental details and related works**

We appreciate all the reviewers’ detailed comments. We apologize that we mistakenly submitted a wrong version with incomplete appendix. After reading the comments, we found that the missing of important details about our algorithms and experiments makes it difficult to fully understand our paper, resulting in some misunderstanding and deviation on the main points of this paper to be evaluated.

As a remedy, a revision with complete appendix has already been uploaded, containing experimental details (Appendix F), some further discussions (e.g., a detailed review of policy representation learning in Appendix D.5). Some parts in the main content are also further clarified to describe the details. We emphasize two major points about our experiments below to correct a few misunderstandings and other minor parts can be found in our updated appendix:
1) We use a normal-scale policy network of 2 layers and 64 units for each (e.g., more than 10k parameters for Ant-v1) in our experiments in Sec. 5, rather than the small policy used in demonstrative experiments in Sec. 3.2. Thus, directly using policy parameters as a representation is infeasible and it motivates us to design effective policy representation.
2) The way of policy update is the same for both PPO and PPO-PeVFA in the experiments. PPO and PPO-PeVFA only differs at value function approximation used for the calculation of GAEs.

We thank Reviewer #2 and #3 for pointing out related works of Policy Evaluation Network (PEN or PVN) [Harb et al., arXiv:2002.11833] and Parameter-based Value Functions (PVFs) [Faccio and Schmidhuber, arXiv:2006.09226]. Similar to our work, they also study value function variants by taking policy as input. However, there are a few key differences between our work and theirs and we have discussed in the revised version in details (related work (in Sec. 2.2) and the review of policy representation learning (in Sec. 4 and Appendix D.5)). Here we summarize the major differences as follows:
1) PVN [Harb et al., arXiv:2002.11833] can be viewed as a predecessor work of Policy-Dynamics Value Function (PDVF) [Raileanu et al., arXiv:2007.02879] which was discussed in Sec. 2.2. Both of them conduct value approximation for a given collection of policies and then optimize policy with gradients through policy-specific inputs (shorted as GTPI below) of well-trained value function variants in zero-shot (offline) manner. We view this as a typical case of global generalization (Figure 2a). In our paper, we focus more on the local generalization scenario and utilize value generalization to improve learning during standard GPI process with no prior policies given. Moreover, in PVN, Network Fingerprint is proposed to represent policy by the concatenation of policy output vectors under a set of optimized probing states. In contrast, we propose a general framework (consists of SPR and OPR) to learn an embedding of RL policy. We provide a detailed discussion on Network Fingerprint in Appendix D.5.
2) PVFs simply take policy parameters (i.e., raw policy representation) as input and utilize value generalization in both online and offline RL. In our paper, policy representation learning is one of our major contributions. This improves the value generalization among policies and allow us to utilize it effectively in standard OpenAI Gym continuous control tasks, rather than the toy environments like CartPole and Swimmer adopted in both PVN and PVFs. Besides, in PVFs, new policy gradients are proposed in the form of a combination of conventional policy gradients plus GTPI (i.e., by backpropagating through policy parameters in PVFs). In our work, we do not use GTPI for the policy update and we utilize value generalization to improve value estimation purely and thus facilitate GPI.
3) Last but not the least, another major contribution of our work compared with PVN and PVFs is that we theoretically analyze the value generalization. In previous works, it is not clear how value generalization among policies can be and whether it is necessarily beneficial to the learning process. We derive the condition of closer approximation in local generalization scenario (Corollary 2) and provide empirical evidence for this (Figure 3b, Figure 8,9). Based on this, we then utilize this potential advantage and propose GPI with PeVFA.


Compared to our first submission, the incremental contents in this revision are colored by orange. More detailed responses to all reviewers’ individual concerns and questions will be added soon.

---

### Author Response · Authors · 2020-11-25
**A Quick View on All the Incremental Contents in the Revision**

In the following, we summarize the main incremental contents of the revision in a quick view:
1) Experimental details are provided in mainly Appendix F.1; experimental analysis on more results (e.g., the sensitivity questioned by Reviewer #1) and visualizations are provided in the remaining parts of Appendix F.
2) A detailed review of previous works that are related to policy representation learning is provided in Appendix D.5. A further discussion of OPR/SPR and CL/AUX can be seen in Appendix D.6.
3) Discussions and comments on related work (i.e., PVN and PVFs, as pointed out by Reviewer #2 and #3) are included in Sec. 2.2 and in the detailed review in Appendix D.5.
4) Figure 9 is added to provide more empirical evidence for the local generalization when larger learning rates are used. A few sentences are added in Sec.3.2 and Appendix B.2 for the ease of reading and understanding Figure 3b and Figure 8,9, as mentioned by Reviewer #1 and #3.
5) Further discussions on off-policy learning, convergence and TD learning are in Appendix C.2.

---

### Decision · Program_Chairs · 2021-01-07
**Final Decision**

**Decision:**

Reject

**Comment:**

This paper proposes to consider value functions as explicit functions of policies, in order to allow generalization not only on the state(action) space, but also on the policy space.
The initial reviews assessed that the paper was dealing with an important RL topic, but also raised many concerns about the position to previous works (PVN and PVF), the theoretical contributions and the experiments. The authors provided a rebuttal and revision that only partly addressed the initial concerns (check also the review of R3, updated to provide additional feedback following the author’s rebuttal).
The final discussion led to the assessment that the paper is not ready for publication. Remaining concerns include clarity, claims being not fully supported by the experiments, theoretical aspects and missing baselines.